# Lower Bounds and Optimal Algorithms for Non-Smooth Convex Decentralized Optimization over Time-Varying Networks

**Dmitry Kovalev**
Yandex Researh
dakovalev1@gmail.com

**Ekaterina Borodich**
MIPT[*]
borodich.ed@phystech.edu

**Alexander Gasnikov**
Innopolis University, MIPT,[*] Skoltech[†]
gasnikov@yandex.ru

**Dmitrii Feoktistov**
Innopolis University,[‡] MSU[§]
feoktistovdd@my.msu.ru

## Abstract

We consider the task of minimizing the sum of convex functions stored in a decentralized manner across the nodes of a communication network. This problem is relatively well-studied in the scenario when the objective functions are smooth, or the links of the network are fixed in time, or both. In particular, lower bounds on the number of decentralized communications and (sub)gradient computations required to solve the problem have been established, along with matching optimal algorithms. However, the remaining and most challenging setting of non-smooth decentralized optimization over time-varying networks is largely underexplored, as neither lower bounds nor optimal algorithms are known in the literature. We resolve this fundamental gap with the following contributions: (i) we establish the first lower bounds on the communication and subgradient computation complexities of solving non-smooth convex decentralized optimization problems over time-varying networks; (ii) we develop the first optimal algorithm that matches these lower bounds and offers substantially improved theoretical performance compared to the existing state of the art.

## 1 Introduction

In this paper, we study the decentralized optimization problem. Specifically, given a set of $n$ compute nodes connected through a communication network, our goal is to solve the following finite-sum optimization problem with quadratic regularization:

$$\min_{x \in \mathbb{R}^d} \left[ p(x) = \frac{1}{n} \sum_{i=1}^{n} f_i(x) + \frac{r}{2} \|x\|^2 \right], \tag{1}$$

where $r \geq 0$ is a regularization parameter, and each function $f_i(x) \colon \mathbb{R}^d \to \mathbb{R}$ is stored on the corresponding node $i \in \{1, \ldots, n\}$. Each node $i$ can perform computations based on its local state and data, and can directly communicate with other nodes through the links in the communication network.

---

[*]Moscow Institute of Physics and Technology

[†]Skolkovo Institute of Science and Technology

[‡]Research Center for Artificial Intelligence, Innopolis University, Innopolis, Russia

[§]Moscow State University

38th Conference on Neural Information Processing Systems (NeurIPS 2024).

Decentralized optimization problems find applications in a wide variety of fields. These include network resource allocation (Beck et al., 2014), distributed model predictive control (Giselsson et al., 2013), power system control (Gan et al., 2012), distributed spectrum sensing (Bazerque and Giannakis, 2009), and optimization in sensor networks (Rabbat and Nowak, 2004). In addition, such problems cover the supervised training of machine learning models through empirical risk minimization, thus attracting significant interest from the machine learning community (Lian et al., 2017; Ryabinin et al., 2021; Ryabinin and Gusev, 2020).

## 1.1 Time-varying Networks

In our paper, we focus on the setting in which the links in the communication network are allowed to change over time. Such time-varying networks (Zadeh, 1961; Kolar et al., 2010) hold significant relevance to many practical applications. For instance, in sensor networks, changes in the link structure can be caused by the motion of sensors and disturbances in the wireless signal connecting pairs of sensors. Similarly, in distributed machine learning, connections between compute nodes can intermittently appear and disappear due to network unreliability (Ryabinin and Gusev, 2020). Lastly, we anticipate that the time-varying setting will be supported by future-generation federated learning systems (Konecnỳ et al., 2016; McMahan et al., 2017), where communication between pairs of mobile devices or between mobile devices and servers will be affected by their physical proximity, which naturally changes over time.

## 1.2 Convex Setting

In this work, we consider the decentralized optimization problem in the case when the objective function is convex (or strongly convex). At first glance, it may seem that the convexity assumption is restrictive and should not be considered. However, as we will see further, even in this fundamental setting, the existing algorithmic developments are limited and have significant gaps that need to be closed. Moreover, considering the convex optimization setting offers important benefits compared to general non-convex functions. One such benefit is that convex optimization often serves as a source of inspiration for the development of algorithms that turn out to be highly effective in solving practical problems, even non-convex ones.

For example, state-of-the-art optimization algorithms such as Adam (Kingma and Ba, 2014) and RMSProp (Hinton et al., 2012) employ the momentum trick, which is observed to be efficient for numerous tasks, including the training of deep neural networks. However, from the perspective of non-convex optimization theory, momentum is useless because, for non-convex problems, the iteration complexity of the standard gradient method cannot be improved (Carmon et al., 2020). On the other hand, it was theoretically proven that momentum substantially boosts the convergence speed of the gradient method when applied to convex functions (Nesterov, 1983). In other words, convex optimization theory suggests that the momentum trick should be used, while non-convex theory suggests that it should not, and the former aligns much more closely with practical observations. A similar situation can be seen with other state-of-the-art optimization methods, including distributed local gradient methods (Mishchenko et al., 2022; Sadiev et al., 2022; Karimireddy et al., 2020), adaptive gradient methods (Duchi et al., 2011), etc. Such inconsistency between non-convex theoretical convergence guarantees for optimization algorithms and their actual performance in practice can be attributed to the fact that the class of non-convex functions is far too broad. This is why many optimization research papers try to narrow down this class by considering additional assumptions such as Polyak-Łojasiewicz condition (Karimi et al., 2016), bounded non-convexity (Carmon et al., 2018; Allen-Zhu, 2018), quasi-strong convexity (Necoara et al., 2019), etc. However, these assumptions can be seen as relaxations of the standard convexity property. Therefore, we naturally opt to focus on the convex decentralized optimization problem, leaving potential generalizations for future work.

## 1.3 Related Work and Main Contributions

Decentralized optimization has been attracting a lot of attention for more than a decade. Plenty of algorithms have been developed, including EXTRA (Shi et al., 2015), DIGing (Nedic et al., 2017), SONATA (Scutari and Sun, 2019), NIDS (Li et al., 2019), APM-C (Li et al., 2018; Rogozin et al., 2021), and many others. In recent years, the focus of the research community has shifted towards the more complex task of finding, in some sense, the best possible algorithms for solving decentralized

Table 1: Summary of the existing state-of-the-art results in decentralized convex optimization. Multiple paper references are provided for each problem setting: papers marked with $*$ provide lower complexity bounds, and papers marked with $\dagger$ provide optimal algorithms that match the corresponding lower bounds.

| | **Smooth Setting** | **Non-Smooth Setting** |
|---|---|---|
| **Fixed Networks** | Kovalev et al. (2020)$^\dagger$ 
 Scaman et al. (2017)$^*$ | Lan et al. (2020)$^\dagger$ 
 Scaman et al. (2018)$^{\dagger*}$ |
| **Time-Varying Networks** | Kovalev et al. (2021a)$^{\dagger*}$ 
 Li and Lin (2021)$^\dagger$ | **Algorithm 1 (this paper)$^\dagger$** 
 **Theorems 1 and 2 (this paper)$^*$** |

optimization problems (Scaman et al., 2017, 2018; Lan et al., 2020; Kovalev et al., 2020, 2021b,a, 2022; Hendrikx et al., 2021; Li et al., 2022; Li and Lin, 2021; Metelev et al., 2024). This task consists of finding a lower bound on the complexity[5] of solving a given subclass of decentralized problems and finding an algorithm whose complexity matches this lower bound. Such algorithms are called optimal because their complexity cannot be improved for a given problem class due to the established lower bounds.

We discuss the four main classes of decentralized optimization problems that cover smooth[6] and non-smooth objective functions, and fixed and time-varying communication networks. We reference the existing state-of-the-art research papers that collectively solve the task of finding optimal algorithms for these classes. These papers are summarized in Table 1. In the case of smooth and strongly convex objective functions and fixed communication networks, Scaman et al. (2017) established the lower bounds on the number of communication rounds and the number of local gradient computations required to find the solution. These lower bounds were matched by OPAPC algorithm of Kovalev et al. (2020). In the case of smooth and strongly convex problems over time-varying networks, lower complexity bounds were provided by Kovalev et al. (2021a), and two optimal algorithms were developed: ADOM+ (Kovalev et al., 2021a) and AccGT (Li and Lin, 2021). In the case of non-smooth convex problems over fixed networks, lower bounds were established by Scaman et al. (2018), and two optimal algorithms were proposed: DCS (Lan et al., 2020) and MSPD (Scaman et al., 2018).

Our paper primarily focuses on the remaining and most challenging setting of non-smooth convex decentralized optimization problems over time-varying networks. Only a few algorithms have been developed for this setting, including the distributed subgradient method (D-SubGD) by Nedic and Ozdaglar (2009), the subgradient-push method (SubGD-Push) by Nedić and Olshevsky (2014), and ZOSADOM by Lobanov et al. (2023). Moreover, to the best of our knowledge, neither lower complexity bounds nor optimal algorithms have been proposed in this setting. Consequently, in this work, we close this significant gap with the following key contributions:

(i) We establish the first lower bounds on the number of decentralized communications and local subgradient computations required to solve problem (1) in the non-smooth convex setting over time-varying networks,

(ii) We show that our lower bounds are tight by developing the first optimal algorithm that matches these lower bounds. The proposed algorithm has state-of-the-art theoretical communication complexity, which outclasses the existing methods described in the literature.

## 2 Notation and Assumptions

In this paper, we are going to use the following notations: $\otimes$ denotes the Kronecker matrix product, $\mathbf{I}_p$ denotes a $p \times p$ identity matrix, $\mathbf{1}_p = (1, \ldots, 1)^\top \in \mathbb{R}^p$, $\mathbf{e}_j^p \in \mathbb{R}^p$ for $j \in \{1, \ldots, p\}$ denotes the $j$-th unit basis vector, where $p \in \{1, 2, \ldots\}$. In addition, $\|\cdot\|$ denotes the standard Euclidean norm of a vector, and $\langle \cdot, \cdot \rangle$ denotes the standard scalar product of two vectors.

---

[5]By complexity, we mean, depending on the context, the number of subgradient computations or decentralized communications required to solve the problem.

[6]A function is called smooth if it is continuously differentiable and has a Lipschitz-continuous gradient.

## 2.1 Objective Function

Further, we describe the assumptions that we impose on problem 1. As discussed in Section 1.2, we assume the convexity of the objective function in problem (1). In particular, we assume that functions $f_1(x), \ldots, f_n(x)$ are convex, which is formally described in Assumption 1.

**Assumption 1.** *Each function $f_i(x)$ is convex. That is, for all $x', x \in \mathbb{R}^d$ and $\tau \in [0, 1]$, the following inequality holds:*

$$f_i(\tau x + (1 - \tau)x') \leq \tau f_i(x) + (1 - \tau)f_i(x'). \qquad (2)$$

In addition, we assume that the objective functions $f_1(x), \ldots, f_n(x)$ are Lipschitz continuous, which is formalized in Assumption 2. This property is widely used in the theoretical analysis of non-smooth optimization algorithms, such as the subgradient method (Nesterov, 2013), dual extrapolation method (Nesterov, 2009), etc.

**Assumption 2.** *Each function $f_i(x)$ is $M$-Lipschitz continuous for $M \geq 0$. That is, for all $x', x \in \mathbb{R}^d$, the following inequality holds:*

$$|f_i(x) - f_i(x')| \leq M\|x - x'\|. \qquad (3)$$

We also need the following Assumption 3, which ensures the existence of a solution to problem (1). Note that in the strongly convex case ($r > 0$), the solution always exists and is unique. However, in the convex case ($r = 0$), we need to explicitly assume the existence of a solution.

**Assumption 3.** *There exists a solution $x^* \in \mathbb{R}^d$ to problem (1) and a distance $R > 0$ such that*

$$\|x^*\| \leq R. \qquad (4)$$

## 2.2 Decentralized Communication

Next, we formally describe the decentralized communication setting. The communication network is typically represented by a graph $\mathcal{G}(\mathcal{V}, \mathcal{E})$, where $\mathcal{V} = \{1, \ldots, n\}$ is the set of compute nodes and $\mathcal{E} \subset \mathcal{V} \times \mathcal{V}$ is the set of links in the network. As mentioned earlier, we allow the communication links to change over time. Thus, we introduce the continuous time parameter $\tau \geq 0$ and a set-valued function $\mathcal{E}(\tau) \colon \mathbb{R}_+ \to 2^{\mathcal{V} \times \mathcal{V}}$, which represents the time-varying set of edges.[7] Our time-varying network is then denoted as $\mathcal{G}(\tau) = (\mathcal{V}, \mathcal{E}(\tau))$.

Decentralized communication is typically represented via a matrix-vector multiplication with the so-called gossip matrix associated with the communication network (Scaman et al., 2017; Kovalev et al., 2021a). In the time-varying setting, we represent the gossip matrix by a matrix-valued function $\mathbf{W}(\tau) \colon \mathbb{R}_+ \to \mathbb{R}^{n \times n}$, which satisfies the following Assumption 4.

**Assumption 4.** *For all $\tau \geq 0$, the gossip matrix $\mathbf{W}(\tau) \in \mathbb{R}^{n \times n}$ associated with the time-varying communication network $\mathcal{G}(\mathcal{V}, \mathcal{E}(\tau))$ satisfies the following properties:*

**(i)** $\mathbf{W}(\tau)_{ij} = 0$ *if* $i \neq j$ *and* $(j, i) \notin \mathcal{E}(\tau)$,

**(ii)** $\mathbf{W}(\tau)\mathbf{1}_n = 0$ *and* $\mathbf{W}(\tau)^\top \mathbf{1}_n = 0$.

We also define the so-called condition number of the network $\chi \geq 1$, which indicates how well the network $\mathcal{G}(\tau)$ is connected (Scaman et al., 2017; Kovalev et al., 2021a). In particular, the communication complexity of most decentralized optimization algorithms depends on $\chi$. Assumption 5 provides the formal definition of this quantity.

**Assumption 5.** *There exists a constant $\chi \geq 1$ such that the following inequality holds for all $\tau \geq 0$:*

$$\|\mathbf{W}(\tau)x - x\|^2 \leq (1 - 1/\chi)\|x\|^2 \text{ for all } x \in \left\{(x_1, \ldots, x_n) \in \mathbb{R}^n : \sum_{i=1}^n x_i = 0\right\}. \qquad (5)$$

# 3 Lower Complexity Bounds

## 3.1 Decentralized Subgradient Optimization Algorithms

In this section, we present the lower bounds on the number of decentralized communications and the number of local subgradient computations required to solve problem (1). These lower bounds

---

[7]By $2^{\mathcal{V} \times \mathcal{V}} = \{\mathcal{E} : \mathcal{E} \subset \mathcal{V} \times \mathcal{V}\}$ we denote the set of all subsets of $\mathcal{V} \times \mathcal{V}$.

apply to a particular class of algorithms, which we refer to as the class of *decentralized subgradient optimization algorithms*. This class can be seen as an adaptation of *black-box optimization procedures* (Scaman et al., 2018) to the time-varying network setting, or an adaptation of *first-order decentralized optimization algorithms* (Kovalev et al., 2021a) to the non-smooth optimization setting.

Non-smooth optimization algorithms typically perform incremental updates by computing the subgradient of a given objective function. The set of all subgradients of a convex function, called the subdifferential, can be multivalued in general. Thus, it is necessary to select the specific subgradient that the algorithm will use. This is done by the *subgradient oracle*, which is described by Definition 1.

**Definition 1.** *For each $i \in \mathcal{V}$, a function $\hat{\nabla} f_i(x) \colon \mathbb{R}^d \to \mathbb{R}^d$ is called a subgradient oracle associated with the function $f_i(x)$ if, for all $x \in \mathbb{R}^d$, it satisfies $\hat{\nabla} f_i(x) \in \partial f_i(x)$. That is, for each $i \in \mathcal{V}$ and for all $x, x' \in \mathbb{R}^d$, the following inequality holds:*

$$f_i(x') \geq f_i(x) + \langle \hat{\nabla} f_i(x), x' - x \rangle. \tag{6}$$

Further, we provide the formal description of the class of decentralized subgradient optimization algorithms in the following Definition 2.

**Definition 2.** *An algorithm is called a decentralized subgradient optimization algorithm with the subgradient computation time $\tau_{sub} > 0$ and decentralized communication time $\tau_{com} > 0$ if it satisfies the following constraints:*

(i) **Internal memory.** *At any time $\tau \geq 0$, each node $i \in \mathcal{V}$ maintains an internal memory, which is represented by a set-valued function $\mathcal{M}_i(\tau) \colon \mathbb{R}_+ \to 2^{\mathbb{R}^d}$. The internal memory can be updated by subgradient computation or decentralized communication, which is formally represented by the following inclusion:*

$$\mathcal{M}_i(\tau) \subset \mathcal{M}_i^{sub}(\tau) \cup \mathcal{M}_i^{com}(\tau), \tag{7}$$

*where set-valued functions $\mathcal{M}_i^{sub}(\tau), \mathcal{M}_i^{com}(\tau) \colon \mathbb{R}_+ \to 2^{\mathbb{R}^d}$ are defined below.*

(ii) **Subgradient computation.** *At any time $\tau \geq 0$, each node $i \in \mathcal{V}$ can update its internal memory $\mathcal{M}_i(\tau)$ by computing the subgradient $\hat{\nabla} f_i(x)$ of the function $f_i(x)$, which takes time $\tau_{sub}$. That is, for all $\tau \geq 0$, the set $\mathcal{M}_i^{sub}(\tau)$ is defined as follows:*

$$\mathcal{M}_i^{sub}(\tau) = \begin{cases} \mathrm{span}(\{x, \hat{\nabla} f_i(x) : x \in \mathcal{M}_i(\tau - \tau_{sub})\}) & \tau \geq \tau_{sub} \\ \varnothing & \tau < \tau_{sub} \end{cases}. \tag{8}$$

(iii) **Decentralized communication.** *At any time $\tau \geq 0$, each node $i \in \mathcal{V}$ can update its internal memory $\mathcal{M}_i(\tau)$ by performing decentralized communication across the communication network, which takes time $\tau_{com}$. That is, for all $\tau \geq 0$, the set $\mathcal{M}_i^{com}(\tau)$ is defined as follows:*

$$\mathcal{M}_i^{com}(\tau) = \begin{cases} \mathrm{span}\big(\bigcup_{(j,i) \in \mathcal{E}(\tau)} \mathcal{M}_j(\tau - \tau_{com})\big) & \tau \geq \tau_{com} \\ \varnothing & \tau < \tau_{com} \end{cases}. \tag{9}$$

(iv) **Initialization and output.** *At time $\tau = 0$, each node $i \in \mathcal{V}$ must initialize its internal memory with the zero vector, that is, $\mathcal{M}_i(0) = \{0\}$. At any time $\tau \geq 0$, each node $i \in \mathcal{V}$ must specify a single output vector from its internal memory, $x_{o,i}(\tau) \in \mathcal{M}_i(\tau)$.*

## 3.2 Lower Bounds

Now, we are ready to present the lower bounds on the execution time $\tau \geq 0$ required to find an $\epsilon$-approximate solution[8] to problem (1) by any algorithm satisfying Definition 2. Theorem 1 provides the lower bound in the strongly convex case ($r > 0$), and Theorem 2 provides the lower bound in the convex case ($r = 0$). These lower bounds naturally depend on the precision $\epsilon > 0$, the parameters of the problem, including the Lipschitz constant $M > 0$, the regularization parameter $r \geq 0$, the distance $R > 0$, and the parameters of the network, including the condition number $\chi \geq 1$, communication time $\tau_{\mathrm{com}} > 0$, and subgradient computation time $\tau_{\mathrm{sub}} > 0$.

---

[8]A vector $x \in \mathbb{R}^d$ is called an $\epsilon$-approximate solution to problem (1) if $p(x) - p(x^*) \leq \epsilon$.

Table 2: Lower bounds on the communication complexity of solving problem (1) in the centralized (Arjevani and Shamir, 2015), decentralized fixed network (Scaman et al., 2018), and decentralized time-varying network (Theorems 1 and 2) settings.

| Setting | Centralized | Fixed networks[9] | Time-varying networks |
|---|---|---|---|
| **Strongly convex** | $\Omega\left(M/\sqrt{r\epsilon}\right)$ | $\Omega\left(\sqrt{\chi}M/\sqrt{r\epsilon}\right)$ | $\Omega\left(\chi M/\sqrt{r\epsilon}\right)$ |
| **Convex** | $\Omega\left(MR/\epsilon\right)$ | $\Omega\left(\sqrt{\chi}MR/\epsilon\right)$ | $\Omega\left(\chi MR/\epsilon\right)$ |

**Theorem 1.** *For arbitrary parameters $M, r, \epsilon, \tau_{com}, \tau_{sub} > 0$ and $\chi \geq 1$, there exists an optimization problem of the form* (1) *satisfying Assumptions 1, 2 and 3, corresponding subgradient oracles given by Definition 1, a time varying network $\mathcal{G}(\tau) = (\mathcal{V}, \mathcal{E}(\tau))$, and a corresponding time-varying gossip matrix $\mathbf{W}(\tau)$ satisfying Assumptions 4 and 5, such that at least the following time $\tau$ is required to reach precision $p(x_{o,i}(\tau)) - p(x^*) \leq \epsilon$ by any decentralized subgradient optimization algorithm satisfying Definition 2:*

$$\tau \geq \Omega\left(\tau_{com} \cdot \frac{\chi M}{\sqrt{r\epsilon}} + \tau_{sub} \cdot \frac{M^2}{r\epsilon}\right). \tag{10}$$

**Theorem 2.** *For arbitrary parameters $M, R, \epsilon, \tau_{com}, \tau_{sub} > 0$ and $\chi \geq 1$, there exists an optimization problem of the form* (1) *with zero regularization ($r = 0$) satisfying Assumptions 1, 2 and 3, corresponding subgradient oracles given by Definition 1, a time varying network $\mathcal{G}(\tau) = (\mathcal{V}, \mathcal{E}(\tau))$, and a corresponding time-varying gossip matrix $\mathbf{W}(\tau)$ satisfying Assumptions 4 and 5, such that at least the following time $\tau$ is required to reach precision $p(x_{o,i}(\tau)) - p(x^*) \leq \epsilon$ by any decentralized subgradient optimization algorithm satisfying Definition 2:*

$$\tau \geq \Omega\left(\tau_{com} \cdot \frac{\chi MR}{\epsilon} + \tau_{sub} \cdot \frac{M^2 R^2}{\epsilon^2}\right). \tag{11}$$

The proofs of Theorems 1 and 2 can be found in Appendix B. Further, we provide a brief and informal description of the main theoretical ideas that underlie these proofs:

(i) We select a specific "hard" instance of problem (1). In particular, we choose the objective function of the form $p(x) = a\sum_{j=1}^{d-1}|\langle \mathbf{e}_{j+1}^d - \mathbf{e}_j^d, x\rangle| - a\langle \mathbf{e}_1^d, x\rangle + \frac{r}{2}\|x\|^2$, which was used by Arjevani and Shamir (2015); Scaman et al. (2018) in the proof of lower bounds on the communication complexity in centralized and fixed-network settings. One can show that the gap $p(x) - p(x^*)$ is lower-bounded by a positive constant as long as the last component of the vector $x$ is zero, and it takes $\Omega(\tau_{\text{sub}} \cdot d)$ time to break this bound due to the constraint on the subgradient updates (8).

(ii) We split the objective function between two nodes of a star-topology network with a time-varying central node, which was previously utilized by Kovalev et al. (2021a) in the proof of lower bounds for optimizing smooth functions. One can show that it takes $\Omega(n) = \Omega(\chi)$ communications to exchange information between the two selected nodes due to the time-varying center. This contrasts with the fixed path-topology network used by Scaman et al. (2017, 2018), where such an exchange would take $\Omega(n) = \Omega(\sqrt{\chi})$ communications. Moreover, using the constraint (8), we can show that it takes $\Omega(\tau_{\text{com}} \cdot nd)$ time to make the last component of the vector $x$ nonzero and break the lower bound on the gap $p(x) - p(x^*)$, thanks to the way we split the objective function.

(iii) Based on the above considerations, we show that the total execution time required to solve the problem is lower-bounded by $\Omega(\tau_{\text{com}} \cdot nd + \tau_{\text{sub}} \cdot d)$. Thus, we obtain the desired results by making a specific choice of the dimension $d$, network size $n$, and other parameters of problem (1).

## 3.3 Comparison with the Lower Bounds in Centralized and Fixed Network Settings

We compare the lower complexity bounds for solving non-smooth convex optimization problems in the three main distributed optimization settings: centralized, decentralized fixed network, and

---

[9]Scaman et al. (2018) do not provide any lower complexity bounds in the strongly convex setting. However, the desired lower bound on the communication complexity can be obtained by extending their analysis.

**Algorithm 1**

1: **input:** $x^0 = x^{-1} = \tilde{x}^0 \in (\mathbb{R}^d)^n$, $y^0 = \overline{y}^0 \in (\mathbb{R}^d)^n$, $z^0 = \overline{z}^0 \in \mathcal{L}^\perp$, $m^0 \in (\mathbb{R}^d)^n$
2: **parameters:** $K, T \in \{1, 2, \ldots\}$, $\{(\alpha_k, \beta_k, \gamma_k, \sigma_k, \lambda_k, \tau_x^k, \eta_x^k, \eta_y^k, \eta_z^k, \theta_z^k)\}_{k=0}^{K-1} \subset \mathbb{R}_+^{10}$
3: **for** $k = 0, 1, \ldots, K-1$ **do**
4: $\quad \underline{y}^k = \alpha_k y^k + (1 - \alpha_k)\overline{y}^k, \quad \underline{z}^k = \alpha_k z^k + (1 - \alpha_k)\overline{z}^k$
5: $\quad g_y^k = \nabla_y G(\underline{y}^k, \underline{z}^k), \quad g_z^k = \nabla_z G(\underline{y}^k, \underline{z}^k), \quad$ where function $G(y, z)$ is defined in eq. (12)
6: $\quad \tilde{g}_z^k = (\mathbf{W}_k \otimes \mathbf{I}_d) g_z^k, \quad \hat{g}_z^k = (\mathbf{W}_k \otimes \mathbf{I}_d)(g_z^k + m^k)$,
$\quad$ where $\mathbf{W}_k$ denotes the gossip matrix $\mathbf{W}(\tau)$ at the current time $\tau$
7: $\quad y^{k+1} = y^k - \eta_y^k(g_y^k + \hat{x}^{k+1}), \quad z^{k+1} = z^k - \eta_z^k \hat{g}_z^k, \quad \hat{x}^{k+1} = x^k + \gamma_k(\tilde{x}^k - x^{k-1})$
8: $\quad \overline{y}^{k+1} = \underline{y}^k + \alpha_k(y^{k+1} - y^k), \quad \overline{z}^{k+1} = \underline{z}^k - \theta_z^k \tilde{g}_z^k, \quad m^{k+1} = (\eta_z^k/\eta_z^{k+1})(m^k + g_z^k - \hat{g}_z^k)$
9: $\quad x^{k,0} = x^{\overline{k}}$
10: $\quad$ **for** $t = 0, 1, \ldots, T-1$ **do**
11: $\quad\quad g_x^{k,t} = (\hat{\nabla} f_1(x_1^{k,t}), \ldots, \hat{\nabla} f_n(x_n^{k,t}))$
12: $\quad\quad x^{k,t+1} = x^{k,t} - \eta_x^k\left(g_x^{k,t} + \beta_k x^{k,t+1} - y^{k+1} + \tau_x^k(x^{k,t+1} - x^k)\right)$
13: $\quad x^{k+1} = \sigma_k x^{k,T} + (1 - \sigma_k)\tilde{x}^{k+1}, \quad \tilde{x}^{k+1} = \frac{1}{T}\sum_{t=1}^T x^{k,t}, \quad \overline{x}^{k+1} = \alpha_k \tilde{x}^{k+1} + (1 - \alpha_k)\overline{x}^k$
14: $(x_a^K, y_a^K, z_a^K) = (\sum_{k=1}^K \lambda_k)^{-1} \sum_{k=1}^K \lambda_k(\overline{x}^k, \overline{y}^k, \overline{z}^k)$
15: **output:** $x_o^K = \frac{1}{n}\sum_{i=1}^n x_{a,i}^K \in \mathbb{R}^d, \quad$ where $(x_{a,1}^K, \ldots, x_{a,n}^K) = x_a^K \in (\mathbb{R}^d)^n$

decentralized time-varying network. The lower subgradient computation complexity bounds coincide in these cases (Nesterov (2013),Scaman et al. (2018),Theorems 1 and 2). However, the situation with the communication complexity is different. See Table 2 for a summary.

Theorems 1 and 2 imply that the communication complexity in the decentralized time-varying network setting is proportional to the network condition number $\chi$. In contrast, the communication complexity in the fixed network setting is proportional to $\sqrt{\chi}$, which reflects the fact that time-varying networks are more difficult to deal with compared to fixed networks. In particular, there was a long-standing conjecture that the "upgrade" from the factor $\chi$ to the factor $\sqrt{\chi}$ in communication complexity is impossible in the time-varying network setting. Only recently, this conjecture was proved for smooth functions by Kovalev et al. (2021a), and now we resolve this open question in the non-smooth case as well.

## 4 Optimal Algorithm

In this section, we develop an optimal algorithm for solving the non-smooth convex decentralized optimization problem (1) over time-varying networks. The design of our algorithm relies on a specific saddle-point reformulation of the problem, which we describe in the following section.

### 4.1 Saddle-Point Reformulation

Let functions $F(x)\colon (\mathbb{R}^d)^n \to \mathbb{R}$ and $G(y, z)\colon (\mathbb{R}^d)^n \times (\mathbb{R}^d)^n \to \mathbb{R}$ be defined as follows:

$$F(x) = \sum_{i=1}^n f_i(x_i) + \frac{r_x}{2}\|x\|^2 \quad \text{and} \quad G(y, z) = \frac{r_{yz}}{2}\|y + z\|^2, \tag{12}$$

where $x = (x_1, \ldots, x_n) \in (\mathbb{R}^d)^n$, and $r_x, r_{yz} > 0$ are some constants that satisfy

$$r_x + 1/r_{yz} = r. \tag{13}$$

Consider the following saddle-point problem:

$$\min_{x \in (\mathbb{R}^d)^n} \max_{y \in (\mathbb{R}^d)^n} \max_{z \in (\mathbb{R}^d)^n} [Q(x, y, z) = F(x) - \langle y, x \rangle - G(y, z)] \quad \text{s.t.} \quad z \in \mathcal{L}^\perp, \tag{14}$$

where $\mathcal{L}^\perp \subset (\mathbb{R}^d)^n$ is the orthogonal complement to the so-called consensus space $\mathcal{L} \subset (\mathbb{R}^d)^n$, defined as follows:

$$\mathcal{L} = \{(x_1, \ldots, x_n) : x_1 = \ldots = x_n\}, \quad \mathcal{L}^\perp = \{(x_1, \ldots, x_n) : \sum_{i=1}^n x_i = 0\}. \tag{15}$$

One can show that the saddle-point problem (14) is equivalent to the minimization problem (1). This is justified by the following Lemma 1. The proof of the lemma can be found in the Appendix A.

**Lemma 1.** *Problem* (14) *is equivalent to problem* (1) *in the following sense:*

$$\min_{x\in(\mathbb{R}^d)^n} \max_{y\in(\mathbb{R}^d)^n} \max_{z\in\mathcal{L}^\perp} Q(x,y,z) = n \cdot \min_{x\in\mathbb{R}^d} p(x). \tag{16}$$

The saddle-point reformulation of the form (14) was first introduced by Kovalev et al. (2020, 2021a) to develop optimal decentralized algorithms for optimizing smooth functions. However, these are not applicable to the non-smooth case. To the best of our knowledge, the only attempt to adapt the reformulation (14) to the non-smooth setting was made by Lobanov et al. (2023). However, their results have significant downsides, which we discuss in Section 4.3.

## 4.2 New Algorithm and its Convergence

Now, we present Algorithm 1 for solving problem (1). We provide upper bounds on the number of decentralized communications $K$ and the number of subgradient computations $K \times T$ required to find an $\epsilon$-approximate solution to the problem. Theorems 3 and 4 provide the upper bounds in the strongly convex ($r > 0$) and convex ($r = 0$) cases, respectively. The proofs can be found in Appendix D. The total execution time of Algorithm 1 is upper-bounded as $\tau = \mathcal{O}\left(\tau_{\text{com}} \cdot K + \tau_{\text{sub}} \cdot K \times T\right)$, where the communication time $\tau_{\text{com}} > 0$ and the subgradient computation time $\tau_{\text{sub}} > 0$ are described in Definition 2. This upper-bound on the execution time cannot be improved because of the lower bounds established in the previous Section 3. Therefore, Algorithm 1 is an optimal algorithm for solving problem (1).

**Theorem 3.** *Under Assumptions 1, 2, 3, 4 and 5, let $r > 0$ (strongly convex case). Then Algorithm 1 requires $K = \mathcal{O}\left(\frac{\chi M}{\sqrt{r\epsilon}}\right)$ decentralized communications (line 6 of Algorithm 1) and $K \times T = \mathcal{O}\left(\frac{M^2}{r\epsilon}\right)$ subgradient computations (line 11 of Algorithm 1) to reach precision $p(x_o^K) - p(x^*) \le \epsilon$.*

**Theorem 4.** *Under Assumptions 1, 2, 3, 4 and 5, let $r = 0$ (convex case). Then Algorithm 1 requires $K = \mathcal{O}\left(\frac{\chi MR}{\epsilon}\right)$ decentralized communications (line 6 of Algorithm 1) and $K \times T = \mathcal{O}\left(\frac{M^2 R^2}{\epsilon^2}\right)$ subgradient computations (line 11 of Algorithm 1) to reach precision $p(x_o^K) - p(x^*) \le \epsilon$.*

The design of Algorithm 1 is based on the fundamental Forward-Backward algorithm (Bauschke and Combettes, 2011). Let $\mathsf{E} = (\mathbb{R}^d)^n \times (\mathbb{R}^d)^n \times \mathcal{L}^\perp$ be a Euclidean space, and consider a monotone operator $A(u)\colon \mathsf{E} \to \mathsf{E}$ and a maximally-monotone multivalued operator $B(u)\colon \mathsf{E} \to 2^{\mathsf{E}}$ defined as follows:

$$A(u) = \begin{bmatrix} 0 \\ \nabla_y G(y,z) \\ \mathbf{P}\nabla_z G(y,z) \end{bmatrix}, \qquad B(u) = \begin{bmatrix} \partial F(x) - y \\ x \\ 0 \end{bmatrix}, \tag{17}$$

where $u = (x,y,z) \in \mathsf{E}$, and $\mathbf{P} = (\mathbf{I}_n - (1/n)\mathbf{1}_n\mathbf{1}_n^\top) \otimes \mathbf{I}_d \in \mathbb{R}^{nd\times nd}$ is the orthogonal projection matrix onto $\mathcal{L}^\perp$. Then problem (14) is equivalent to the following monotone inclusion problem:

$$\text{find } u \in \mathsf{E} \text{ such that } 0 \in A(u) + B(u). \tag{18}$$

The basic Forward-Backward algorithm iterates $u^{k+1} = (\text{id} + B)^{-1}(u^k - A(u^k))$, where id is the identity operator and $(\text{id} + B)^{-1}$ denotes the inverse of the operator $\text{id}(u) + B(u)$, which is called resolvent. Algorithm 1 can be obtained by making the following major modifications to these iterations:

  **(i)** We accelerate the convergence of the Forward-Backward algorithm using Nesterov acceleration (Nesterov, 1983). Although this mechanism cannot be applied to the general monotone inclusion problem (18), Kovalev et al. (2020) showed that it can be used when the operator $A(u)$ is equal to the gradient of a smooth convex function, which is true in our case.

  **(ii)** Computation of the operator $A(u)$ requires multiplication with the matrix $\mathbf{P}$. This, in turn, requires an exact averaging of a vector, which is difficult to do over the time-varying network. Kovalev et al. (2021b) showed that this obstacle can be tackled with the Error-Feedback mechanism for decentralized communication, which we also utilize.

  **(iii)** At each iteration of the algorithm, we have to compute the resolvent, which requires solving an auxiliary subproblem $\min_x \max_y \frac{\tau_x}{2}\|x - x^k\|^2 + F(x) - \langle y, x\rangle - \frac{\tau_y}{2}\|y - y^k\|^2$. This problem cannot be solved exactly, so we have to find an approximate solution using an

Table 3: The execution time $\tau$ required to find an $\epsilon$-approximate solution to the decentralized optimization problem (1) by the following algorithms: D-SubGD (Nedic and Ozdaglar, 2009), SubGD-Push (Nedić and Olshevsky, 2014), ZO-SADOM (Lobanov et al., 2023), and Algorithm 1 (this paper). Decentralized communication and subgradient computation complexities are marked with green and yellow colors, respectively. For D-SubGD, the complexity is not provided because the algorithm converges only to a neighborhood of the solution. For SubGD-Push, $\text{poly}(M,R,d)$ denotes a certain polynomial in $M, R, d$. For ZO-SADOM, the differences from the optimal complexities are highlighted in red color.

| Algorithm | Strongly-convex case complexity | Convex case complexity |
|---|---|---|
| D-SubGD | N/A ||
| SubGD-Push | $\tau_{\text{com}} \cdot \dfrac{\text{poly}(M,R,d) \cdot n^{2n} \log^2 \frac{1}{\epsilon}}{\epsilon^2} + \tau_{\text{sub}} \cdot \dfrac{\text{poly}(M,R,d) \cdot n^{2n} \log^2 \frac{1}{\epsilon}}{\epsilon^2}$ ||
| ZO-SADOM | $\tau_{\text{com}} \cdot \dfrac{\chi M d^{1/4} \log \frac{1}{\epsilon}}{\sqrt{r\epsilon}} + \tau_{\text{sub}} \cdot \dfrac{M^2 d \log \frac{1}{\epsilon}}{r\epsilon}$ | $\tau_{\text{com}} \cdot \dfrac{\chi M R d^{1/4} \log \frac{1}{\epsilon}}{\epsilon} + \tau_{\text{sub}} \cdot \dfrac{M^2 R^2 d \log \frac{1}{\epsilon}}{\epsilon^2}$ |
| **Algorithm 1** | $\tau_{\text{com}} \cdot \dfrac{\chi M}{\sqrt{r\epsilon}} + \tau_{\text{sub}} \cdot \dfrac{M^2}{r\epsilon}$ | $\tau_{\text{com}} \cdot \dfrac{\chi M R}{\epsilon} + \tau_{\text{sub}} \cdot \dfrac{M^2 R^2}{\epsilon^2}$ |
| **Lower Bounds** | $\tau_{\text{com}} \cdot \dfrac{\chi M}{\sqrt{r\epsilon}} + \tau_{\text{sub}} \cdot \dfrac{M^2}{r\epsilon}$ | $\tau_{\text{com}} \cdot \dfrac{\chi M R}{\epsilon} + \tau_{\text{sub}} \cdot \dfrac{M^2 R^2}{\epsilon^2}$ |

additional "inner" algorithm based on the subgradient method (Nesterov, 2013) and the Chambolle-Pock operator splitting (Chambolle and Pock, 2011). We also have to conduct a careful analysis to find an efficient way to combine the inner and the "outer" Forward-Backward algorithms and avoid unnecessary waste of subgradient calls.

The design of Algorithm 1 shares some similarities with the algorithm of Kovalev et al. (2021a) such as **(i)** and **(ii)** above. However, Kovalev et al. (2021a) simply add the gradient $\nabla F(x)$ to the operator $A(u)$ and use the accelerated version of the Forward-Backward algorithm, which we obviously cannot do as the function $F(x)$ is not smooth. Instead, we have to put the subdifferential $\partial F(x)$ into the operator $B(u)$ and follow **(iii)** above. Part **(iii)**, in turn, shares some similarities with the algorithm of Lan et al. (2020). However, Lan et al. (2020) simply have a zero operator $A(u) = 0$, which makes **(i)** and **(ii)** above unnecessary in their case. In contrast, we cannot make such simplifications because we work in the much more complicated setting of time-varying networks.

### 4.3 Comparison with the Existing Results

One could naturally expect that the existing optimal algorithms, originally developed for fixed networks, such as DCS (Lan et al., 2020) and MSPD (Scaman et al., 2018), could be applied to solve problem (1) over time-varying networks. However, this is not the case, which is justified by the lack of corresponding theoretical guarantees and was shown empirically by Kovalev et al. (2021b). Therefore, we have to consider only those algorithms that were specifically developed for the time-varying network setting.

We provide a comparison of our Algorithm 1 with the existing state-of-the-art decentralized methods for solving convex non-smooth optimization problems over time-varying networks in Table 3.[10] These include D-SubGD (Nedic and Ozdaglar, 2009), SubGD-Push (Nedić and Olshevsky, 2014), and ZO-SADOM (Lobanov et al., 2023). The first two algorithms have poor performance: D-SubGD converges only to limited precision, and SubGD-Push converges at a slow rate of $\mathcal{O}(\log^2(1/\epsilon)/\epsilon^2)$, which does not match even the iteration complexity of the standard centralized subgradient method, let alone the improved complexity of Algorithm 1. The complexity of ZO-SADOM is also worse than the lower bounds. Moreover, the theoretical results of Lobanov et al. (2023) have substantial drawbacks compared to ours:

---

[10]We ignore universal constants in Table 3 like in the $\mathcal{O}(\cdot)$ and $\Omega(\cdot)$ notation.

**(i)** Lobanov et al. (2023) do not provide any theoretical insights or innovations in the analysis of their algorithm. In particular, they use the randomized smoothing technique (Duchi et al., 2012) to obtain a smooth approximation of the objective $p(x)$, and apply the existing algorithm of Kovalev et al. (2021a) to minimize this approximation. In contrast, we develop a new algorithm that directly works with the original non-smooth objective $p(x)$.

**(ii)** ZO-SADOM has extra factors $d^{1/4} \log(1/\epsilon)$ and $d \log(1/\epsilon)$ in the decentralized communication and subgradient computation complexities, respectively, compared to the optimal complexity of our Algorithm 1. Thus, the performance of ZO-SADOM can be poor when applied, for instance, to large-scale machine learning problems in which the dimension $d$ can be huge.

## Acknowledgments and Disclosure of Funding

This research has been financially supported by The Analytical Center for the Government of the Russian Federation (Agreement No. 70-2021-00143 01.11.2021, IGK 000000D730324P540002).

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

# Appendix

## A Proof of Lemma 1

The orthogonal complement $\mathcal{L}^\perp$ to the consensus space $\mathcal{L}$ is given as follows:

$$\mathcal{L}^\perp = \left\{ (x_1, \ldots, x_n) \in (\mathbb{R}^d)^n : x_1 + \ldots + x_n = 0 \right\}. \tag{19}$$

Let us perform the maximization of $Q(x, y, z)$ in the variable $y \in (\mathbb{R}^d)^n$:

$$\max_{y \in (\mathbb{R}^d)^n} Q(x, y, z) \stackrel{(a)}{=} \max_{y \in (\mathbb{R}^d)^n} F(x) + \langle y, x \rangle - G(y, z)$$

$$\stackrel{(b)}{=} F(x) + \max_{y \in (\mathbb{R}^d)^n} \left[ \langle y, x \rangle - \frac{r_{yz}}{2} \|y + z\|^2 \right]$$

$$= F(x) + \frac{1}{2r_{yz}} \|x\|^2 - \langle x, z \rangle,$$

where (a) uses the definition of $Q(x, y, z)$ eq. (14); (b) uses the definition of $G(y, z)$ in eq. (12). Next, we perform maximization in the variable $z \in \mathcal{L}^\perp$:

$$\max_{z \in \mathcal{L}^\perp} \max_{y \in (\mathbb{R}^d)^n} Q(x, y, z) = \max_{z \in \mathcal{L}^\perp} \left[ F(x) + \frac{1}{2r_{yz}} \|x\|^2 - \langle x, z \rangle \right]$$

$$= F(x) + \frac{1}{2r_{yz}} \|x\|^2 + \max_{z \in \mathcal{L}^\perp} \left[ -\langle x, z \rangle \right]$$

$$= F(x) + \frac{1}{2r_{yz}} \|x\|^2 + I_{\mathcal{L}}(x),$$

where $I_{\mathcal{L}}(x) \colon (\mathbb{R}^d)^n \to \mathbb{R}$ is the indicator function, which is defined as follows:

$$I_{\mathcal{L}}(x) = \max_{z \in \mathcal{L}^\perp} \left[ -\langle x, z \rangle \right] = \begin{cases} 0 & x \in \mathcal{L} \\ +\infty & \text{otherwise} \end{cases}. \tag{20}$$

Now, we can rewrite the saddle-point problem (14) as follows

$$\min_{x \in (\mathbb{R}^d)^n} \max_{y \in (\mathbb{R}^d)^n} \max_{z \in \mathcal{L}^\perp} Q(x, y, z) \stackrel{(a)}{=} \min_{x \in (\mathbb{R}^d)^n} \max_{z \in \mathcal{L}^\perp} \max_{y \in (\mathbb{R}^d)^n} Q(x, y, z)$$

$$\stackrel{(b)}{=} \min_{x \in (\mathbb{R}^d)^n} F(x) + \frac{1}{2r_{yz}} \|x\|^2 + I_{\mathcal{L}}(x)$$

$$\stackrel{(c)}{=} \min_{x \in (\mathbb{R}^d)^n} \sum_{i=1}^n \left( f_i(x_i) + \frac{r_x + 1/r_{yz}}{2} \|x_i\|^2 \right) + I_{\mathcal{L}}(x)$$

$$\stackrel{(d)}{=} \min_{x \in (\mathbb{R}^d)^n} \sum_{i=1}^n \left( f_i(x_i) + \frac{r}{2} \|x_i\|^2 \right) + I_{\mathcal{L}}(x)$$

$$\stackrel{(e)}{=} n \cdot \min_{x \in \mathbb{R}^d} p(x).$$

where (a) uses the fact that we can exhange the order of the two consecutive maximizations; (b) uses the previous equation; (c) uses the definition of $F(x)$ in eq. (12); (d) uses eq. (13); (e) uses the definition of $p(x)$ in eq. (1) and the definition of $I_{\mathcal{L}}(x)$. $\square$

# B  Proof of Theorems 1 and 2

## B.1  The Hard Instance of Problem (1)

**Compute nodes.**   In this proof, we consider the case when $\chi \geq 3$. The case $\chi < 3$ can be proven using the fixed-network argument of Scaman et al. (2018). We choose $n = 3\lfloor \chi/3 \rfloor$, which implies that $n \geq 3$ and $n \bmod 3 = 0$. We also divide the set of nodes $\mathcal{V} = \{1, \ldots, n\}$ into the following three disjoint subsets: $\mathcal{V}_1 = \{1, \ldots, n/3\}$, $\mathcal{V}_2 = \{n/3 + 1, \ldots, 2n/3\}$ and $\mathcal{V}_3 = \{2n/3 + 1, \ldots, n\}$.

**Objective functions.**   We fix an arbitrary odd integer $d \in \{3, 5, 7, \ldots\}$ and define functions $f_1(x), \ldots, f_n(x) \colon \mathbb{R}^d \to \mathbb{R}$ as follows:

$$f_i(x) = \begin{cases} a\sum_{j=1}^{(d-1)/2} h_{2j-1}(x) - a\langle x, \mathbf{e}_1^d \rangle & i \in \mathcal{V}_1 \\ a\sum_{j=1}^{(d-1)/2} h_{2j}(x) & i \in \mathcal{V}_2 \\ 0 & i \in \mathcal{V}_3 \end{cases}, \tag{21}$$

where $a > 0$ is an arbitrary constant and functions $h_1(x), \ldots, h_{d-1}(x) \colon \mathbb{R}^d \to \mathbb{R}$ are defined as follows:

$$h_j(x) = \left| \langle x, \mathbf{e}_{j+1}^d - \mathbf{e}_j^d \rangle \right|. \tag{22}$$

Consequently, the objective function $p(x)$ in problem (1) is given as follows:

$$p(x) = \frac{a}{3} \sum_{j=1}^{d-1} h_j(x) - \frac{a}{3} \langle \mathbf{e}_1^d, x \rangle + \frac{r}{2} \|x\|^2. \tag{23}$$

We also define the subgradient oracles $\hat{\nabla} f_1(x), \ldots, \hat{\nabla} f_n(x) \colon \mathbb{R}^d \to \mathbb{R}^d$ as follows:

$$\hat{\nabla} f_i(x) = \begin{cases} a\sum_{j=1}^{(d-1)/2} \hat{\nabla} h_{2j-1}(x) - a\mathbf{e}_1^d & i \in \mathcal{V}_1 \\ a\sum_{j=1}^{(d-1)/2} \hat{\nabla} h_{2j}(x) & i \in \mathcal{V}_2 \\ 0 & i \in \mathcal{V}_3 \end{cases}, \tag{24}$$

where $\hat{\nabla} h_1(x), \ldots, \hat{\nabla} h_{d-1}(x) \colon \mathbb{R}^d \to \mathbb{R}^d$ are the subgradient oracles associated with functions $h_1(x), \ldots, h_{d-1}(x)$, defined as follows:

$$\hat{\nabla} h_j(x) = \begin{cases} \mathbf{e}_{j+1}^d - \mathbf{e}_j^d & \langle \mathbf{e}_{j+1}^d, x \rangle > \langle \mathbf{e}_j^d, x \rangle \\ 0 & \langle \mathbf{e}_{j+1}^d, x \rangle = \langle \mathbf{e}_j^d, x \rangle \\ \mathbf{e}_j^d - \mathbf{e}_{j+1}^d & \langle \mathbf{e}_{j+1}^d, x \rangle < \langle \mathbf{e}_j^d, x \rangle \end{cases}. \tag{25}$$

**Time-varying network.**   We choose the time-varying network $\mathcal{G}(\tau) = (\mathcal{V}, \mathcal{E}(\tau))$ to be a star-topology undirected graph with the time-varying center node $i_c(\tau) \in \mathcal{V}$. Formally, we define the edges of the time-varying network $\mathcal{E}(\tau) \subset \mathcal{V} \times \mathcal{V}$ as follows:

$$\mathcal{E}(\tau) = \bigcup_{i \in \mathcal{V}, i \neq i_c(\tau)} \{(i, i_c(\tau)), (i_c(\tau), i)\}. \tag{26}$$

We also specify the center node $i_c(\tau)$ at a given time $\tau \geq 0$ as follows:

$$i_c(\tau) = 2n/3 + 1 + \left( \lfloor \tau/\tau_{\mathrm{com}} \rfloor \bmod n/3 \right). \tag{27}$$

We choose the time-varying gossip matrix $\mathbf{W}(\tau) \in \mathbb{R}^{n \times n}$ to be the Laplacian matrix of the graph $\mathcal{G}(\tau)$. Formally, $\mathbf{W}(\tau)$ is defined as follows:

$$\mathbf{W}(\tau)_{ij} = \frac{1}{n} \begin{cases} 0 & i \neq j \text{ and } (i, j) \notin \mathcal{E}(\tau) \\ -1 & i \neq j \text{ and } (i, j) \in \mathcal{E}(\tau) \\ \deg_i(\tau) & i = j \end{cases}, \tag{28}$$

where $\deg_i(\tau)$ denotes the degree of the node $i \in \mathcal{V}$ in the graph $\mathcal{G}(\tau)$, i.e.,

$$\deg_i(\tau) = |\{j : (i, j) \in \mathcal{E}(\tau)\}|. \tag{29}$$

One can observe, that the time-varying gossip matrix $\mathbf{W}(\tau)$ satisfies Assumption 4, in particular, $\ker \mathbf{W}(\tau) = \ker \mathbf{W}(\tau)^\top = \mathrm{span}(\{\mathbf{1}_n\})$. Moreover, one can show that $\mathbf{W}(\tau)$ is a symmetric matrix, and $\lambda_{\max}(\mathbf{W}(\tau)) = 1$ and $\lambda_{\min}^+(\mathbf{W}(\tau)) = 1/n \geq 1/\chi$. Hence, $\mathbf{W}(\tau)$ satisfies Assumption 5.

## B.2 Auxiliary Lemmas

Further, we define linear spaces $\mathcal{K}_0, \ldots, \mathcal{K}_d \subset \mathbb{R}^d$ as follows:

$$\mathcal{K}_0 = \{0\} \quad \text{and} \quad \mathcal{K}_j = \operatorname{span}\left(\{\mathbf{e}_1^d, \ldots, \mathbf{e}_j^d\}\right) \quad \text{for} \quad j \in \{1, \ldots, d\}. \tag{30}$$

In order to prove Theorems 1 and 2, we will use the following auxiliary lemmas. The proofs of these lemmas can be found in Appendix C. Furthermore, the proof of Theorem 1 is contained in Appendix B.3, and the proof of Theorem 2 is contained in Appendix B.4.

**Lemma 2.** *For all $\tau \geq 0$, the following statements hold:*

    **(i)** *Let $i \in \mathcal{V}_1$. Then, for all $j \in \{1, \ldots, (d-1)/2\}$,*

$$\mathcal{M}_i(\tau) \subset \mathcal{K}_{2j} \quad \text{implies} \quad \mathcal{M}_i^{sub}(\tau + \tau_{sub}) \subset \mathcal{K}_{2j}. \tag{31}$$

    **(ii)** *Let $i \in \mathcal{V}_2$. Then, for all $j \in \{0, \ldots, (d-1)/2\}$,*

$$\mathcal{M}_i(\tau) \subset \mathcal{K}_{2j+1} \quad \text{implies} \quad \mathcal{M}_i^{sub}(\tau + \tau_{sub}) \subset \mathcal{K}_{2j+1}. \tag{32}$$

    **(ii)** *Let $i \in \mathcal{V}_3$. Then, for all $j \in \{0, \ldots, d\}$,*

$$\mathcal{M}_i(\tau) \subset \mathcal{K}_j \quad \text{implies} \quad \mathcal{M}_i^{sub}(\tau + \tau_{sub}) \subset \mathcal{K}_j. \tag{33}$$

The proof of Lemma 2 is contained in Appendix C.1.

**Lemma 3.** *Let $k \in \{0, \ldots, n(d-1)/6 - 1\}$. Then, for all $\tau < (k+1)\tau_{com}$, the following inclusion holds:*

$$\mathcal{M}_i(\tau) \subset \begin{cases} \mathcal{K}_{2p+2} & i \in \mathcal{V}_1 \text{ or } (i \in \mathcal{V}_3 \text{ and } i \leq 2n/3 + q + 1) \\ \mathcal{K}_{2p+1} & i \in \mathcal{V}_2 \text{ or } (i \in \mathcal{V}_3 \text{ and } i > 2n/3 + q + 1) \end{cases}, \tag{34}$$

*where $p = \lfloor 3k/n \rfloor$ and $q = k \bmod (n/3)$.*

The proof of Lemma 3 is contained in Appendix C.2.

**Lemma 4.** *Let functions $f_1, \ldots, f_n(x)$ be defined by eq. (21). Then problem eq. (1) has a unique solution $x^* \in \mathbb{R}^d$, which is given as follows:*

$$x^* = \frac{a}{3rd}\mathbf{1}_d. \tag{35}$$

*Moreover, for all $x \in \mathcal{K}_{d-1}$, the following inequality holds:*

$$p(x) - p(x^*) \geq \frac{a^2}{18rd}. \tag{36}$$

The proof of Lemma 4 is contained in Appendix C.3.

## B.3 Proof of Theorem 1

**Decentralized communication.** Lemma 3 implies that $\mathcal{M}_i(\tau) \subset \mathcal{K}_{d-1}$ as long as $\tau < \tau_{com} \cdot n(d-1)/6$. Hence, Lemma 4 implies eq. (36) for all $x \in \mathcal{M}_i(\tau)$ as long as $\tau < \tau_{com} \cdot n(d-1)/6$. Let the constant $a > 0$ be chosen as follows:

$$a = \frac{M}{2\sqrt{d}}. \tag{37}$$

Then, each function $f_i(x)$ defined by eq. (21) is $M$-Lipschitz. Indeed, the case $i \in \mathcal{V}_3$ is trivial. In the case when $i \in \mathcal{V}_1$, we can prove the $M$-Lipschitz continuity of $f_i(x)$ as follows:

$$\begin{aligned} f_i(x) - f_i(x') &= a \sum_{j=1}^{(d-1)/2} \left(\left|\langle x, \mathbf{e}_{2j}^d - \mathbf{e}_{2j-1}^d\rangle\right| - \left|\langle x', \mathbf{e}_{2j}^d - \mathbf{e}_{2j-1}^d\rangle\right|\right) - a\langle x - x', \mathbf{e}_1^d\rangle \\ &\leq a \sum_{j=1}^{(d-1)/2} \left|\langle x - x', \mathbf{e}_{2j}^d - \mathbf{e}_{2j-1}^d\rangle\right| + a\left|\langle x - x', \mathbf{e}_1^d\rangle\right| \end{aligned}$$

$$\leq a \sum_{j=1}^{(d-1)/2} \left( |\langle x - x', \mathbf{e}_{2j}^d \rangle| + |\langle x - x', \mathbf{e}_{2j-1}^d \rangle| \right) + a |\langle x - x', \mathbf{e}_1^d \rangle|$$

$$= a \sum_{j=1}^{d-1} |\langle x - x', \mathbf{e}_j^d \rangle| + a |\langle x - x', \mathbf{e}_1^d \rangle|$$

$$\leq 2a \sum_{j=1}^{d} |\langle x - x', \mathbf{e}_j^d \rangle| \leq 2a\sqrt{d}\|x - x'\| \leq M\|x - x'\|.$$

In the case when $i \in \mathcal{V}_2$, we can prove the $M$-Lipschitz continuity of $f_i(x)$ similarly.

Without loss of generality, we assume $\epsilon \leq M^2/(576r)$ and define $d \in \{3, 5, \ldots\}$ as follows:

$$d = 2\left\lfloor \frac{M}{12\sqrt{r\epsilon}} \right\rfloor - 1. \tag{38}$$

Using eqs. (37) and (38), for all $\tau < \tau_{\text{com}} \cdot n(d-1)/6$ and $x \in \mathcal{M}_i(\tau)$, we obtain

$$p(x) - p(x^*) \geq \frac{M^2}{36rd^2} > \epsilon.$$

Hence, to reach precision $p(x) - p(x^*) \leq \epsilon$ for some $x \in \mathcal{M}_i(\tau)$, it is necessary that $\tau$ satisfies

$$
\begin{aligned}
\tau &\geq \tau_{\text{com}} \cdot \frac{n(d-1)}{6} \\
&= \tau_{\text{com}} \cdot \left\lfloor \frac{\chi}{3} \right\rfloor \left( \left\lfloor \frac{M}{12\sqrt{r\epsilon}} \right\rfloor - 1 \right) \\
&\geq \tau_{\text{com}} \cdot \frac{\chi}{3} \left( \frac{M}{12\sqrt{r\epsilon}} - 1 \right) \\
&= \Omega\left( \tau_{\text{com}} \cdot \frac{M\chi}{\sqrt{r\epsilon}} \right).
\end{aligned} \tag{39}
$$

**Subgradient computation.** We also need to prove that to reach precision $p(x) - p(x^*) \leq \epsilon$ for some $x \in \mathcal{M}_i(\tau)$, it is necessary that $\tau$ satisfies

$$\tau \geq \Omega\left( \tau_{\text{sub}} \cdot \frac{M^2}{r\epsilon} \right). \tag{40}$$

We can do this by providing an extended version of our hard problem instance, described in Appendix B.1. In particular, we consider the following instance of problem (1):

$$\min_{(x,x') \in \mathbb{R}^d \times \mathbb{R}^{d'}} \frac{1}{n} \sum_{i=1}^{n} (f_i(x) + f_i'(x')) + \frac{r}{2}\|x\|^2 + \frac{r}{2}\|x'\|^2, \tag{41}$$

where functions $f_1(x), \ldots, f_n(x) \colon \mathbb{R}^d \to \mathbb{R}$ are defined in Appendix B.1 by eq. (21), and functions $f_1'(x'), \ldots, f_n'(x') \colon \mathbb{R}^{d'} \to \mathbb{R}$ are defined as follows:

$$f_i'(x') = b \max_{j \in \{1, \ldots, d'\}} \langle \mathbf{e}_j^{d'}, x' \rangle, \tag{42}$$

where $b > 0$ is some constant. Then, by choosing an appropriate subgradient oracle $\hat{\nabla} f_i'(x')$ associated with each function $f_i'(x')$ (see Section 3.2.1 of Nesterov (2013)) we can obtain both lower bounds (39) and (40), which concludes the proof. $\qquad\square$

## B.4 Proof of Theorem 2

Our proof of Theorem 2 is very similar to the proof of Theorem 1 with the following differences. Let function $h_\delta(x) \colon \mathbb{R}^d \to \mathbb{R}$ be the Huber function, which is defined as follows:

$$h_\delta(x) = \sum_{j=1}^{d} h_\delta^j(\langle \mathbf{e}_j^d, x \rangle), \quad \text{where} \quad h_\delta^j(t) = \begin{cases} \frac{1}{2}t^2 & |t| \leq \delta \\ \delta|t| - \frac{1}{2}\delta^2 & |t| > \delta \end{cases}. \tag{43}$$

Note that function $h_\delta(x)$ is continuously differentiable and $(\sqrt{d}\delta)$-Lipschitz continuous.

In the proof of Theorem 3 we used functions $f_1(x), \ldots, f_n(x)$ defined in eq. (21) of Appendix B.1. Here we use a slightly different choice, that is, functions $f_1(x), \ldots, f_n(x)$ are defined as follows:

$$f_i(x) = h_\delta(x) + \begin{cases} a\sum_{j=1}^{(d-1)/2} h_{2j-1}(x) - a\langle x, \mathbf{e}_1^d \rangle & i \in \mathcal{V}_1 \\ a\sum_{j=1}^{(d-1)/2} h_{2j}(x) & i \in \mathcal{V}_2 \\ 0 & i \in \mathcal{V}_3 \end{cases}. \tag{44}$$

Consequently, our hard instance of problem (1), which is described in Appendix B.1, turns into the following:

$$\min_{x \in \mathbb{R}^d} \left[ p(x) = \frac{a}{3} \sum_{j=1}^{d-1} h_j(x) - \frac{a}{3}\langle \mathbf{e}_1^d, x \rangle + ch_\delta(x) \right], \tag{45}$$

where $c > 0$ is some constant, and functions $h_1(x), \ldots, h_{d-1}(x)$ are defined in eq. (22).

One can show that Lemmas 2 and 3 still hold true. We can also replace Lemma 4 with the following Lemma 5. The proof of this lemma is a trivial extension of the proof of Lemma 4, which uses the fact that $\nabla(\frac{1}{2}\|\cdot\|^2)(x^*) = \nabla h_\delta(x^*)$ as long as $\delta$ and $x^*$ are defined by eq. (46) and eq. (47), respectively.

**Lemma 5.** *Let $\delta$ be defined as follows:*

$$\delta = \frac{a}{3cd}. \tag{46}$$

*Problem eq. (45) has a solution $x^* \in \mathbb{R}^d$, which is given as follows:*

$$x^* = \frac{a}{3cd}\mathbf{1}_d. \tag{47}$$

*Moreover, for all $x \in \mathcal{K}_{d-1}$, the following inequality holds:*

$$p(x) - p(x^*) \geq \frac{a^2}{18cd}. \tag{48}$$

One can also show that each function $f_i(x)$ defined in eq. (44) is $M_f$-Lipschitz continuous, where $M_f$ is defined as follows:

$$M_f = 2a\sqrt{d} + c\delta\sqrt{d} = 2a\sqrt{d} + a/(3\sqrt{d}) \leq 3a\sqrt{d}. \tag{49}$$

Let us choose $a$ and $c$ as follows:

$$a = \frac{M}{3\sqrt{d}} \quad \text{and} \quad c = \frac{M}{9Rd}. \tag{50}$$

This choice of $a$ and $c$ implies $M_f \leq M$ and $\|x^*\| \leq R$. Moreover, eq. (48) implies

$$p(x) - p(x^*) \geq \frac{MR}{18d} \tag{51}$$

as long as $x^* \in \mathcal{K}_{d-1}$. Next, without loss of generality we can assume $\epsilon \leq (MR)/72$ and choose $d \in \{3, 5, \ldots\}$ as follows:

$$d = 2\left\lfloor \frac{MR}{36\epsilon} \right\rfloor - 1, \tag{52}$$

which, for all $x \in \mathcal{M}_i(\tau)$, implies

$$p(x) - p(x^*) > \epsilon$$

as long as $\tau$ satisfies

$$\tau \geq \tau_{\text{com}} \cdot \frac{n(d-1)}{6} = \Omega\left(\tau_{\text{com}} \cdot \frac{MR\chi}{\epsilon}\right), \tag{53}$$

which concludes the proof. $\qquad\square$

# C Proofs of Lemmas from Section B.2

## C.1 Proof of Lemma 2

**Statement (i).** Let $i \in \mathcal{V}_1$ and $x \subset \mathcal{K}_{2j}$ for $j \in \{1, \ldots, (d-1)/2\}$. Then for $l \geq 2j+1$ we obtain $\langle \mathbf{e}_{l+1}^d - \mathbf{e}_l^d, x \rangle = 0$, which implies $\hat{\nabla} h_l(x) = 0$ due to eq. (25). Hence, we obtain the following:

$$
\begin{aligned}
\frac{1}{a} \hat{\nabla} f_i(x) &\overset{(a)}{=} \hat{\nabla} h_1(x) + \hat{\nabla} h_3(x) + \cdots + \hat{\nabla} h_{d-2}(x) - \mathbf{e}_1^d \\
&\overset{(b)}{=} \hat{\nabla} h_1(x) + \hat{\nabla} h_3(x) + \cdots + \hat{\nabla} h_{2j-1}(x) - \mathbf{e}_1^d \\
&\overset{(c)}{\subset} \operatorname{span}\left(\{\mathbf{e}_1^d, \mathbf{e}_2^d\} \cup \cdots \cup \{\mathbf{e}_{2j-1}^d, \mathbf{e}_{2j}^d\}\right) \\
&\overset{(d)}{\subset} \mathcal{K}_{2j},
\end{aligned}
$$

where (a) uses eq. (24); (b) uses the fact that $\hat{\nabla} h_l(x) = 0$ for $l \geq 2j+1$; (c) uses eq. (25); (d) uses the definition of $\mathcal{K}_{2j}$ in eq. (30). Hence, $\mathcal{M}_i(\tau) \subset \mathcal{K}_{2j}$ implies $\mathcal{M}_i^{\text{sub}}(\tau + \tau_{\text{sub}}) \subset \mathcal{K}_{2j}$ by the definition of $\mathcal{M}_i^{\text{sub}}(\cdot)$ in eq. (8).

**Statement (ii).** Let $i \in \mathcal{V}_2$ and $x \subset \mathcal{K}_{2j+1}$ for $j \in \{0, \ldots, (d-1)/2\}$. Then for $l \geq 2j+2$ we obtain $\langle \mathbf{e}_{l+1}^d - \mathbf{e}_l^d, x \rangle = 0$, which implies $\hat{\nabla} h_l(x) = 0$ due to eq. (25). Hence, we obtain the following:

$$
\begin{aligned}
\frac{1}{a} \hat{\nabla} f_i(x) &\overset{(a)}{=} \hat{\nabla} h_2(x) + \hat{\nabla} h_4(x) + \cdots + \hat{\nabla} h_{d-1}(x) \\
&\overset{(b)}{=} \hat{\nabla} h_2(x) + \hat{\nabla} h_4(x) + \cdots + \hat{\nabla} h_{2j}(x) \\
&\overset{(c)}{\subset} \operatorname{span}\left(\{\mathbf{e}_2^d, \mathbf{e}_3^d\} \cup \cdots \cup \{\mathbf{e}_{2j}^d, \mathbf{e}_{2j+1}^d\}\right) \\
&\overset{(d)}{\subset} \mathcal{K}_{2j},
\end{aligned}
$$

where (a) uses eq. (24); (b) uses the fact that $\hat{\nabla} h_l(x) = 0$ for $l \geq 2j+2$; (c) uses eq. (25); (d) uses the definition of $\mathcal{K}_{2j+1}$ in eq. (30). Hence, $\mathcal{M}_i(\tau) \subset \mathcal{K}_{2j+1}$ implies $\mathcal{M}_i^{\text{sub}}(\tau + \tau_{\text{sub}}) \subset \mathcal{K}_{2j+1}$ by the definition of $\mathcal{M}_i^{\text{sub}}(\cdot)$ in eq. (8).

**Statement (iii).** This statement is trivially implied by the definition of $\hat{\nabla} f_i(x)$ in eq. (24) and the definition of $\mathcal{M}_i^{\text{sub}}(\cdot)$ in eq. (8). $\qquad\square$

## C.2 Proof of Lemma 3

We prove the lemma using the induction on $k$.

**Base case: $k = 0$.** In this case, we assume $\tau < (k+1)\tau_{\text{com}} = \tau_{\text{com}}$. Hence, for all $i \in \mathcal{V}$, we obtain $\mathcal{M}_i^{\text{com}}(\tau) = \varnothing$ and $\mathcal{M}_i(\tau) \subset \mathcal{M}_i^{\text{sub}}(\tau)$. Using Lemma 2 and the fact that $\mathcal{M}_i^{\text{com}}(\tau) = \varnothing$, we can easily obtain

$$
\mathcal{M}_i(\tau) \subset \mathcal{M}_i^{\text{sub}}(\tau) \subset \begin{cases} \mathcal{K}_2 & i \in \mathcal{V}_1 \\ \mathcal{K}_1 & i \in \mathcal{V}_2 \\ \mathcal{K}_0 & i \in \mathcal{V}_3 \end{cases},
$$

which implies the desired eq. (34) for $k = p = q = 0$.

**Induction hypothesis.** Let $k' \in \{0, 1, 2, \ldots\}$. We assume that eq. (34) holds for all $\tau < (k' + 1)\tau_{\text{com}}$, that is,

$$
\mathcal{M}_i(\tau) \subset \begin{cases} \mathcal{K}_{2p'+2} & i \in \mathcal{V}_1 \text{ or } (i \in \mathcal{V}_3 \text{ and } i \leq 2n/3 + q' + 1) \\ \mathcal{K}_{2p'+1} & i \in \mathcal{V}_2 \text{ or } (i \in \mathcal{V}_3 \text{ and } i > 2n/3 + q' + 1) \end{cases}, \tag{54}
$$

where $p' = \lfloor 3k'/n \rfloor$ and $q' = k' \bmod (n/3)$.

**Induction step.** We assume that the induction hypothesis (54) is true. Our goal is to prove that eq. (34) holds for $k = k' + 1$. When $0 \leq \tau < k\tau_{\text{com}}$, the desired eq. (34) is implied by the induction hypothesis (54). Thus, we can assume $k\tau_{\text{com}} \leq \tau < (k+1)\tau_{\text{com}}$. Further, we consider two cases: $q \neq 0$ and $q = 0$.

**Induction step, case $q \neq 0$.** In this case, $p = p'$ and $q = q' + 1$.

**Part (i).** First, we consider the case

$$k\tau_{\text{com}} \leq \tau < \min\{(k+1)\tau_{\text{com}}, k\tau_{\text{com}} + \tau_{\text{sub}}\}. \tag{55}$$

Equation (55) implies $\tau - \tau_{\text{sub}} < (k'+1)\tau_{\text{com}}$ and $\tau - \tau_{\text{com}} < (k'+1)\tau_{\text{com}}$. Using the induction hypothesis (54) and the fact that $p' = p$ and $q' = q - 1$, we get

$$\mathcal{M}_i(\tau - \tau_{\text{sub}}), \ \mathcal{M}_i(\tau - \tau_{\text{com}}) \subset \begin{cases} \mathcal{K}_{2p+2} & i \in \mathcal{V}_1 \text{ or } (i \in \mathcal{V}_3 \text{ and } i \leq 2n/3 + q) \\ \mathcal{K}_{2p+1} & i \in \mathcal{V}_2 \text{ or } (i \in \mathcal{V}_3 \text{ and } i > 2n/3 + q) \end{cases}. \tag{56}$$

Hence, using Lemma 2, we obtain

$$\mathcal{M}_i^{\text{sub}}(\tau) \subset \begin{cases} \mathcal{K}_{2p+2} & i \in \mathcal{V}_1 \text{ or } (i \in \mathcal{V}_3 \text{ and } i \leq 2n/3 + q) \\ \mathcal{K}_{2p+1} & i \in \mathcal{V}_2 \text{ or } (i \in \mathcal{V}_3 \text{ and } i > 2n/3 + q) \end{cases}. \tag{57}$$

Equations (27) and (55) imply $i_c(\tau) = 2n/3 + q + 1$. Hence, using eq. (56), we get

$$\mathcal{M}_{i_c(\tau)}(\tau - \tau_{\text{com}}) \subset \mathcal{K}_{2p+1}.$$

For $i \neq i_c(\tau)$, using eqs. (9) and (56), we get

$$\mathcal{M}_i^{\text{com}}(\tau) = \text{span}\left(\mathcal{M}_{i_c(\tau)}(\tau - \tau_{\text{com}})\right) \subset \mathcal{K}_{2p+1}. \tag{58}$$

For $i = i_c(\tau) = 2n/3 + q + 1$, using eqs. (9) and (56), we get

$$\mathcal{M}_{i_c(\tau)}^{\text{com}}(\tau) = \text{span}\left(\bigcup_{j \neq i_c(\tau)} \mathcal{M}_j(\tau - \tau_{\text{com}})\right) \subset \mathcal{K}_{2p+2}. \tag{59}$$

Hence, using eqs. (58) and (59), for all $i \in \mathcal{V}$, we obtain

$$\mathcal{M}_i^{\text{com}}(\tau) \subset \begin{cases} \mathcal{K}_{2p+2} & i = 2n/3 + q + 1 \\ \mathcal{K}_{2p+1} & i \neq 2n/3 + q + 1 \end{cases}. \tag{60}$$

Now, we combine eqs. (57) and (60), and obtain

$$\mathcal{M}_i(\tau) \subset \mathcal{M}_i^{\text{sub}}(\tau) \cup \mathcal{M}_i^{\text{com}}(\tau) \subset \begin{cases} \mathcal{K}_{2p+2} & i \in \mathcal{V}_1 \text{ or } (i \in \mathcal{V}_3 \text{ and } i \leq 2n/3 + q + 1) \\ \mathcal{K}_{2p+1} & i \in \mathcal{V}_2 \text{ or } (i \in \mathcal{V}_3 \text{ and } i > 2n/3 + q + 1) \end{cases}. \tag{61}$$

Thus, we were able to prove eq. (34) for $\tau$ satisfying (55).

**Part (ii).** We can prove the general case

$$k\tau_{\text{com}} \leq \tau < \min\{(k+1)\tau_{\text{com}}, k\tau_{\text{com}} + l\tau_{\text{sub}}\}$$

for arbitrary $l \in \{1, 2, \ldots\}$ using the induction on $l$. The only difference compared to the proof in the previous part is in eq. (56), which will change to

$$\mathcal{M}_i(\tau - \tau_{\text{sub}}) \subset \begin{cases} \mathcal{K}_{2p+2} & i \in \mathcal{V}_1 \text{ or } (i \in \mathcal{V}_3 \text{ and } i \leq 2n/3 + q + 1) \\ \mathcal{K}_{2p+1} & i \in \mathcal{V}_2 \text{ or } (i \in \mathcal{V}_3 \text{ and } i > 2n/3 + q + 1) \end{cases},$$

and eq. (57) will change as follows due to Lemma 2:

$$\mathcal{M}_i^{\text{sub}}(\tau) \subset \begin{cases} \mathcal{K}_{2p+2} & i \in \mathcal{V}_1 \text{ or } (i \in \mathcal{V}_3 \text{ and } i \leq 2n/3 + q + 1) \\ \mathcal{K}_{2p+1} & i \in \mathcal{V}_2 \text{ or } (i \in \mathcal{V}_3 \text{ and } i > 2n/3 + q + 1) \end{cases}.$$

However, the rest of the proof, including eq. (61) will remain unchanged.

**Induction step, case $q = 0$.** In this case $p = p' + 1$ and $q' = n/3 - 1$.

**Part (i).** First, we consider the case

$$k\tau_{\text{com}} \leq \tau < \min\{(k+1)\tau_{\text{com}}, k\tau_{\text{com}} + \tau_{\text{sub}}\}. \tag{62}$$

Equation (62) implies $\tau - \tau_{\text{sub}} < (k'+1)\tau_{\text{com}}$ and $\tau - \tau_{\text{com}} < (k'+1)\tau_{\text{com}}$. Using the induction hypothesis (54) and the fact that $p' = p - 1$ and $q' = n/3 - 1$, we get

$$\mathcal{M}_i(\tau - \tau_{\text{sub}}), \ \mathcal{M}_i(\tau - \tau_{\text{com}}) \subset \begin{cases} \mathcal{K}_{2p} & i \in \mathcal{V}_1 \text{ or } i \in \mathcal{V}_3 \\ \mathcal{K}_{2p-1} & i \in \mathcal{V}_2 \end{cases}. \tag{63}$$

Equations (27) and (62) imply $i_c(\tau) = 2n/3 + 1$. Using eq. (63), we get

$$\mathcal{M}_{i_c(\tau)}(\tau - \tau_{\text{com}}) \subset \mathcal{K}_{2p}.$$

For $i \neq i_c(\tau)$, using eqs. (9) and (63), we get

$$\mathcal{M}_i^{\text{com}}(\tau) = \text{span}\left(\mathcal{M}_{i_c(\tau)}(\tau - \tau_{\text{com}})\right) \subset \mathcal{K}_{2p}. \tag{64}$$

For $i = i_c(\tau) = 2n/3 + 1$, using eqs. (9) and (63), we get

$$\mathcal{M}_{i_c(\tau)}^{\text{com}}(\tau) = \text{span}\left(\bigcup_{j \neq i_c(\tau)} \mathcal{M}_j(\tau - \tau_{\text{com}})\right) \subset \mathcal{K}_{2p}. \tag{65}$$

Hence, using eqs. (64) and (65), for all $i \in \mathcal{V}$, we obtain

$$\mathcal{M}_i^{\text{com}}(\tau) \subset \mathcal{K}_{2p}. \tag{66}$$

Using Lemma 2, from eq. (63) we obtain

$$\mathcal{M}_i^{\text{sub}}(\tau) \subset \begin{cases} \mathcal{K}_{2p} & i \in \mathcal{V}_1 \text{ or } i \in \mathcal{V}_3 \\ \mathcal{K}_{2p-1} & i \in \mathcal{V}_2 \end{cases}. \tag{67}$$

Hence, using eqs. (66) and (67), for all $i \in \mathcal{V}$, we obtain

$$\mathcal{M}_i(\tau) \subset \mathcal{M}_i^{\text{sub}}(\tau) \cup \mathcal{M}_i^{\text{com}}(\tau) \subset \mathcal{K}_{2p}, \tag{68}$$

which implies eq. (34) for $\tau$ satisfying (62).

**Part (ii).** Next, we consider the case

$$k\tau_{\text{com}} + \tau_{\text{sub}} \leq \tau < \min\{(k+1)\tau_{\text{com}}, k\tau_{\text{com}} + 2\tau_{\text{sub}}\}. \tag{69}$$

Equation (66) still holds for all $i \in \mathcal{V}$ and $\tau$ satisfying eq. (69). From eqs. (68) and (69), for all $i \in \mathcal{V}$, we obtain

$$\mathcal{M}_i(\tau - \tau_{\text{sub}}) \subset \mathcal{K}_{2p},$$

which, due to Lemma 2, implies the following:

$$\mathcal{M}_i^{\text{sub}}(\tau) \subset \begin{cases} \mathcal{K}_{2p} & i \in \mathcal{V}_1 \text{ or } i \in \mathcal{V}_3 \\ \mathcal{K}_{2p+1} & i \in \mathcal{V}_2 \end{cases}. \tag{70}$$

Hence, using eqs. (66) and (70), we obtain

$$\mathcal{M}_i(\tau) \subset \mathcal{M}_i^{\text{sub}}(\tau) \cup \mathcal{M}_i^{\text{com}}(\tau) \subset \begin{cases} \mathcal{K}_{2p} & i \in \mathcal{V}_1 \text{ or } i \in \mathcal{V}_3 \\ \mathcal{K}_{2p+1} & i \in \mathcal{V}_2 \end{cases}, \tag{71}$$

which implies eq. (34) for $\tau$ satisfying (69).

**Part(iii).** We can prove the general case

$$k\tau_{\text{com}} + l\tau_{\text{sub}} \leq \tau < \min\{(k+1)\tau_{\text{com}}, k\tau_{\text{com}} + (l+1)\tau_{\text{sub}}\} \tag{72}$$

for $l \in \{2, 3, \ldots\}$ using the induction on $l$. There will be no differences compared to the proof in the previous part. Indeed, eqs. (66) and (71) will still hold for all $i \in \mathcal{V}$ and $\tau$ satisfying eq. (72). $\quad\square$

## C.3 Proof Lemma 4

One can show, that $x^*$ defined in eq. (35) is indeed the unique minimizer of the function $p(x)$ defined in eq. (23). Moreover, we can obtain the following:

$$p(x^*) = -\frac{a^2}{18rd}.$$

We can lower-bound function $p(x)$ as follows:

$$p(x) = \frac{r}{2}\|x\|^2 - \frac{a}{3}\langle \mathbf{e}_1^d, x\rangle + \frac{a}{3}\sum_{j=1}^{d-1}|\langle x, \mathbf{e}_{j+1}^d - \mathbf{e}_j^d\rangle|$$

$$\geq -\frac{a}{3}|\langle \mathbf{e}_1^d, x\rangle| + \frac{a}{3}\sum_{j=1}^{d-1}\left(|\langle x, \mathbf{e}_j^d\rangle| - |\langle x, \mathbf{e}_{j+1}^d\rangle|\right)$$

$$= -\frac{a}{3}|\langle \mathbf{e}_d^d, x\rangle|$$

$$= 0$$

as long as $x \in \mathcal{K}_{d-1}$. Hence, for all $x \in \mathcal{K}_j$, we obtain

$$p(x) - p(x^*) \geq \frac{a^2}{18rd},$$

which concludes the proof. $\qquad\square$

# D Proof of Theorems 3 and 4

## D.1 Auxiliary Lemmas

In order to prove Theorems 3 and 4, we will use the following auxiliary lemmas. The proofs of these lemmas can be found in Appendix E. Furthermore, the proof of Theorem 3 is contained in Appendix D.2, and the proof of Theorem 4 is contained in Appendix D.3.

**Lemma 6.** *Under Assumptions 1, 2 and 3, let $r > 0$ (strongly convex case). Then there exists a solution $(w^*, y^*, z^*) \in \mathcal{L} \times (\mathbb{R}^d)^n \times \mathcal{L}^\perp$ to problem (1), which satisfies the following conditions*

$$0 \in \partial_x Q(w^*, y^*, z^*), \quad 0 = \nabla_y Q(w^*, y^*, z^*), \quad \mathcal{L} \ni \nabla_z Q(w^*, y^*, z^*). \tag{73}$$

*Moreover, the following inequalities hold:*

$$\|w^*\|^2 \le nM^2/r^2, \quad \|y^*\|^2 \le (1 + r_x/r)^2 nM^2, \quad \|z^*\|^2 \le 4nM^2. \tag{74}$$

The proof of Lemma 6 is contained in Appendix E.1.

**Lemma 7.** *Under Assumptions 1 and 2, let $\eta_x^0, \ldots, \eta_x^{K-1}$ and $\beta_0, \ldots, \beta_{K-1}$ be chosen as follows:*

$$\eta_x^k = 1/(\tau_x^k T), \quad \beta_k = r_x, \quad \sigma_k = \tau_x^k/(2\tau_x^k + \beta_k) \quad for \quad k \in \{0, \ldots, K-1\}. \tag{75}$$

*Then, for all $x \in (\mathbb{R}^d)^n$ and $k \in \{0, \ldots, K-1\}$, the following inequality holds:*

$$(\tau_x^k + \tfrac{1}{2}r_x)\|x^{k+1} - x\|^2 \le \tau_x^k \|x^k - x\|^2 + 2nM^2/(\tau_x^k T) \tag{76}$$
$$- \left( F(\tilde{x}^{k+1}) - F(x) - \langle y^{k+1}, \tilde{x}^{k+1} - x \rangle + \tfrac{1}{2}\tau_x^k \|\tilde{x}^{k+1} - x^k\|^2 \right).$$

The proof of Lemma 7 is contained in Appendix E.2.

**Lemma 8.** *Under Assumption 4, for all $k \in \{0, \ldots, K-1\}$, the iterates of Algorithm 1 satisfy*

$$\mathbf{P}z^k = z^k, \quad \mathbf{P}\bar{z}^{k+1} = \bar{z}^{k+1}, \quad \mathbf{P}\underline{z}^k = \underline{z}^k, \tag{77}$$

*where $\mathbf{P} \in \mathbb{R}^{nd \times nd}$ is the orthogonal projection matrix onto $\mathcal{L}^\perp$, which is given as follows:*

$$\mathbf{P} = (\mathbf{I}_n - \tfrac{1}{n}\mathbf{1}_n\mathbf{1}_n^\top) \otimes \mathbf{I}_d. \tag{78}$$

The proof of Lemma 8 is contained in Appendix E.3.

**Lemma 9.** *Under Assumptions 4 and 5, for all $k \in \{0, \ldots, K-1\}$ the following inequality holds:*

$$\|\eta_z^k m^k\|_\mathbf{P}^2 \le 2\chi \|\eta_z^k m^k\|_\mathbf{P}^2 - 2\chi \|\eta_z^{k+1} m^{k+1}\|_\mathbf{P}^2 + 4\chi^2 \|\eta_z^k g_z^k\|_\mathbf{P}^2. \tag{79}$$

The proof of Lemma 9 is contained in Appendix E.4.

**Lemma 10.** *Under Assumptions 4 and 5, let parameters $\theta_z^0, \ldots, \theta_z^{K-1}$ be chosen as follows:*

$$\theta_z^k = 1/(2r_{yz}) \quad for \quad k = 0, \ldots, K-1. \tag{80}$$

*Then, for all $k \in \{0, \ldots, K-1\}$, the following inequality holds:*

$$0 \le -\alpha_k^{-1} \left( \langle \bar{z}^{k+1} - \underline{z}^k, g_z^k \rangle + r_{yz} \|\bar{z}^{k+1} - \underline{z}^k\|^2 \right) - (4\alpha_k \chi r_{yz})^{-1} \|g_z^k\|_\mathbf{P}^2. \tag{81}$$

The proof of Lemma 10 is contained in Appendix E.5.

**Lemma 11.** *Under Assumptions 1 and 2 and under conditions of Lemmas 7 and 10, let parameters $\alpha_0, \ldots, \alpha_{K-1}$ and $\gamma_0, \ldots, \gamma_{K-1}$ be chosen as follows:*

$$\alpha_k = 3/(k+3), \quad \gamma_k = (k+2)/(k+3) \quad for \quad k = 0, \ldots, K-1. \tag{82}$$

*Let parameters $\tau_x^0, \ldots, \tau_x^{K-1}, \eta_y^0, \ldots, \eta_y^{K-1}$, and $\eta_z^0, \ldots, \eta_z^{K-1}$ be chosen as follows:*

$$\tau_x^k = \tau_x \alpha_k^{-1}, \quad \eta_y^k = \eta_y \alpha_k^{-1}, \quad \eta_z^k = \eta_z \alpha_k^{-1} \quad for \quad k = 0, \ldots, K-1, \tag{83}$$

*where $\tau_x, \eta_y$ and $\eta_z$ are defined as follows:*

$$\tau_x = \tfrac{1}{2}r_x, \quad \eta_y = (4r_{yz})^{-1}, \quad \eta_z = (10r_{yz}\chi^2)^{-1}, \quad r_x = \tfrac{2}{3}r, \quad r_{yz} = 3/r. \tag{84}$$

*Let parameters $\lambda_1, \ldots, \lambda_K$ be chosen as follows:*

$$\lambda_K = \alpha_{K-1}^{-2} \quad and \quad \lambda_k = \alpha_{k-1}^{-2} + \alpha_k^{-1} - \alpha_k^{-2} \quad for \quad k = 1, \ldots, K-1. \tag{85}$$

*Let the input of Algorithm 1 be chosen as follows:*

$$x^0 = 0, \quad y^0 = 0, \quad z^0 = 0, \quad m^0 = 0. \tag{86}$$

*Then, for all $x, y \in (\mathbb{R}^d)^n$ and $z \in \mathcal{L}^\perp$, the following inequality holds:*

$$Q(x_a^K, y, z) - Q(x, y_a^K, z_a^K) \le \frac{2}{K^2}\left( r\|x\|^2 + \frac{18}{r}\|y\|^2 + \frac{45\chi^2}{r}\|z\|^2 \right) + \frac{72nM^2}{rKT}. \tag{87}$$

The proof of Lemma 11 is contained in Appendix E.6.

## D.2 Proof of Theorem 3

We can upper-bound $\frac{r_x}{2}\|x_a^K - w^*\|^2$, where $w^*$ is defined in Lemma 6, as follows:

$$
\begin{aligned}
\frac{r_x}{2}\|x_a^K - w^*\|^2 &\overset{(a)}{\leq} Q(x_a^K, y^*, z^*) - Q(w^*, y^*, z^*) \\
&\overset{(b)}{\leq} Q(x_a^K, y^*, z^*) - Q(w^*, y_a^K, z_a^K) \\
&\overset{(c)}{\leq} \frac{2}{K^2}\left(r\|w^*\|^2 + \frac{18}{r}\|y^*\|^2 + \frac{45\chi^2}{r}\|z^*\|^2\right) + \frac{72nM^2}{rKT} \\
&\overset{(d)}{\leq} \frac{2}{K^2}\left(\frac{nM^2}{r} + \frac{18(1+r_x/r)^2nM^2}{r} + \frac{180n\chi^2M^2}{r}\right) + \frac{72nM^2}{rKT}
\end{aligned}
$$

where (a) uses Lemma 6 and the strong convexity of $Q(x, y, z)$ in $x$; (b) and (d) use Lemma 6; (c) uses Lemma 11. Using the definition of $r_x$ in eq. (84)

$$
\begin{aligned}
r\|x_a^K - w^*\|^2 &\leq \frac{6}{K^2}\left(\frac{51nM^2}{r} + \frac{180n\chi^2M^2}{r}\right) + \frac{72nM^2}{rKT} \\
&\leq \frac{1386n\chi^2M^2}{rK^2} + \frac{72nM^2}{rKT}.
\end{aligned}
$$

Next, we can upper-bound $n(p(x_o^K) - p(x^*))$ as follows:

$$
\begin{aligned}
n(p(x_o^K) - p(x^*)) &\overset{(a)}{=} \sum_{i=1}^{n}\left(f_i(x_o^K) - f_i(x^*) + \frac{r}{2}\|x_o^K\|^2 - \frac{r}{2}\|x^*\|^2\right) \\
&\overset{(b)}{=} \sum_{i=1}^{n}\left(f_i(x_o^K) - f_i(x^*) + \frac{r}{2}\|\tfrac{1}{n}\sum_{j=1}^{n} x_{a,j}^K\|^2 - \frac{r}{2}\|x^*\|^2\right) \\
&\overset{(c)}{\leq} \sum_{i=1}^{n}\left(f_i(x_o^K) - f_i(x^*) + \frac{r}{2}\|x_{a,i}^K\|^2 - \frac{r}{2}\|x^*\|^2\right) \\
&\overset{(d)}{=} \sum_{i=1}^{n}\left(f_i(x_o^K) - f_i(x^*)\right) + \frac{r}{2}\|x_a^K\|^2 - \frac{r}{2}\|w^*\|^2 \\
&\overset{(e)}{\leq} \sum_{i=1}^{n}\left(f_i(x_{a,i}^K) - f_i(x^*) + M\|x_{a,i}^K - x_o^K\|\right) + \frac{r}{2}\|x_a^K\|^2 - \frac{r}{2}\|w^*\|^2 \\
&\overset{(f)}{=} F(x_a^K) - F(w^*) + \frac{1}{2r_{yz}}\|x_a^K\|^2 - \frac{1}{2r_{yz}}\|w^*\|^2 + \sum_{i=1}^{n} M\|x_{a,i}^K - x_o^K\| \\
&\overset{(g)}{\leq} F(x_a^K) - F(w^*) + \frac{1}{2r_{yz}}\|x_a^K\|^2 - \frac{1}{2r_{yz}}\|w^*\|^2 \\
&\quad + \sqrt{\textstyle\sum_{i=1}^{n} M^2}\sqrt{\textstyle\sum_{i=1}^{n}\|x_{a,i}^K - x_o^K\|^2} \\
&\overset{(h)}{=} F(x_a^K) - F(w^*) + \frac{1}{2r_{yz}}\|x_a^K\|^2 - \frac{1}{2r_{yz}}\|w^*\|^2 + \sqrt{n}M\|x_a^K\|_{\mathbf{P}}
\end{aligned}
$$

where (a) uses the definition of $p(x)$ in eq. (1); (b) uses the definition of $x_o^K$ on line 15 of Algorithm 1; (c) uses the convexity of $\|\cdot\|^2$; (d) uses the definition of $w^*$ in eq. (95); (e) uses Assumption 2; (f) uses the definition of function $F(x)$ in eq. (12) and eq. (13); (g) uses the Cauchy-Schwarz inequality; (h) uses the definition of $\mathbf{P}$ in eq. (78).

Next, for arbitrary $z \in \mathcal{L}^\perp$ we define $y = -r_{yz}^{-1}x_a^K - z$. Then, we get $\nabla_y Q(x_a^K, y, z) = 0$ and $Q(x_a^K, y, z) = F(x_a^K) + \frac{1}{2r_{yz}}\|x_a^K\|^2 + \langle x_a^K, z\rangle$. Plugging this into the previous upper-bound gives the following:

$$
n(p(x_o^K) - p(x^*)) \leq Q(x_a^K, y, z) - F(w^*) - \frac{1}{2r_{yz}}\|w^*\|^2 - \langle x_a^K, z\rangle + \sqrt{n}M\|x_a^K\|_{\mathbf{P}}
$$

$$\stackrel{(a)}{=} Q(x_a^K, y, z) - Q(w^*, y^*, z^*) - \langle x_a^K, z \rangle + \sqrt{n}M\|x_a^K\|_{\mathbf{P}}$$

$$\stackrel{(b)}{\leq} Q(x_a^K, y, z) - Q(w^*, y_a^K, z_a^K) - \langle x_a^K, z \rangle + \sqrt{n}M\|x_a^K\|_{\mathbf{P}}$$

where (a) uses the definition of $y^*$ in eq. (98) and the definition of $z^*$ in eq. (99); (b) uses Lemma 6.
Next, we choose $z \in \mathcal{L}^{\perp}$ as follows:

$$z = \begin{cases} +\sqrt{n}M\|\mathbf{P}x_a^K\|^{-1}\mathbf{P}x_a^K & x_a^K \neq 0 \\ 0 & x_a^K = 0 \end{cases}. \tag{88}$$

Then, $\langle x_a^K, z \rangle = +\sqrt{n}M\|x_a^K\|_{\mathbf{P}}$ and we obtain the following:

$$n(p(x_o^K) - p(x^*))$$

$$\leq Q(x_a^K, y, z) - Q(w^*, y_a^K, z_a^K)$$

$$\stackrel{(a)}{\leq} \frac{2}{K^2}\left(r\|w^*\|^2 + \frac{18}{r}\|y\|^2 + \frac{45\chi^2}{r}\|z\|^2\right) + \frac{72nM^2}{rKT}$$

$$\stackrel{(b)}{=} \frac{2}{K^2}\left(r\|w^*\|^2 + \frac{18}{r}\|r_{yz}^{-1}x_a^K + z\|^2 + \frac{45\chi^2}{r}\|z\|^2\right) + \frac{72nM^2}{rKT}$$

$$= \frac{2}{K^2}\left(r\|w^*\|^2 + \frac{18}{r}\|r_{yz}^{-1}(x_a^K - w^* + w^*) + z\|^2 + \frac{45\chi^2}{r}\|z\|^2\right) + \frac{72nM^2}{rKT}$$

$$\stackrel{(c)}{\leq} \frac{2}{K^2}\left(r\|w^*\|^2 + \frac{54}{rr_{yz}^2}\|x_a^K - w^*\|^2 + \frac{54}{rr_{yz}^2}\|w^*\|^2 + \frac{54}{r}\|z\|^2 + \frac{45\chi^2}{r}\|z\|^2\right) + \frac{72nM^2}{rKT}$$

$$\leq \frac{2}{K^2}\left(r\|w^*\|^2 + \frac{54}{rr_{yz}^2}\|x_a^K - w^*\|^2 + \frac{54}{rr_{yz}^2}\|w^*\|^2 + \frac{99\chi^2}{r}\|z\|^2\right) + \frac{72nM^2}{rKT}$$

$$\stackrel{(d)}{\leq} \frac{2}{K^2}\left(r\|w^*\|^2 + \frac{54}{rr_{yz}^2}\|x_a^K - w^*\|^2 + \frac{54}{rr_{yz}^2}\|w^*\|^2 + \frac{99n\chi^2M^2}{r}\right) + \frac{72nM^2}{rKT}$$

$$\stackrel{(e)}{\leq} \frac{2}{K^2}\left(\frac{nM^2}{r} + \frac{54nM^2}{r^3r_{yz}^2} + \frac{54}{rr_{yz}^2}\|x_a^K - w^*\|^2 + \frac{99n\chi^2M^2}{r}\right) + \frac{72nM^2}{rKT}$$

$$\stackrel{(f)}{=} \frac{2}{K^2}\left(\frac{7nM^2}{r} + 6r\|x_a^K - w^*\|^2 + \frac{99n\chi^2M^2}{r}\right) + \frac{72nM^2}{rKT}$$

$$\leq \frac{212n\chi^2M^2}{rK^2} + \frac{72nM^2}{rKT} + \frac{12r}{K^2}\|x_a^K - w^*\|^2$$

$$\stackrel{(g)}{\leq} \frac{212n\chi^2M^2}{rK^2} + \frac{72nM^2}{rKT} + \frac{12}{K^2}\left(\frac{1386n\chi^2M^2}{rK^2} + \frac{72nM^2}{rKT}\right),$$

where (a) uses Lemma 11; (b) uses our choice of $y$; (c) uses the parallelogram rule and Young's inequality; (d) uses our choice of $z$; (e) uses Lemma 6; (f) uses the definition of $r_{yz}$ in eq. (84); (g) uses the previously obtained upper-bound on $r\|x_a^K - w^*\|^2$. Dividing both sides of the inequality by $n$ gives the following:

$$p(x_o^K) - p(x^*) \leq \frac{212\chi^2M^2}{rK^2} + \frac{72M^2}{rKT} + \frac{12}{K^2}\left(\frac{1386\chi^2M^2}{rK^2} + \frac{72M^2}{rKT}\right).$$

Hence, choosing the parameters $K$ and $T$ such that

$$K \geq \mathcal{O}\left(\frac{\chi M}{\sqrt{r}\epsilon}\right) \quad \text{and} \quad K \times T \geq \mathcal{O}\left(\frac{M^2}{r\epsilon}\right)$$

implies $p(x_o^K) - p(x^*) \leq \epsilon$, which concludes the proof. $\qquad \square$

### D.3 Proof of Theorem 4

With $r = 0$, the original problem (1) turns into the following problem:

$$\min_{x \in \mathbb{R}^d} \left[ \bar{f}(x) = \frac{1}{n} \sum_{i=1}^{n} f_i(x) \right]. \tag{89}$$

Let $x^* \in \mathbb{R}^d$ be the solution to problem (89), such that $\|x^*\| \leq R$, which always exists due to Assumption 3. Let $r > 0$ be an arbitrary regularization parameter. We can upper-bound function $\bar{f}(x)$ using the regularized objective function $p(x)$ defined in eq. (1) as follows:

$$\bar{f}(x) \leq \bar{f}(x) + \frac{r}{2}\|x\|^2 = p(x).$$

On the other hand, we can lower-bound $\bar{f}(x^*)$ as follows:

$$\bar{f}(x^*) = p(x^*) - \frac{r}{2}\|x^*\|^2 \geq \min_{x' \in \mathbb{R}^d} p(x') - \frac{r}{2}\|x^*\|^2 \geq \min_{x' \in \mathbb{R}^d} p(x') - \frac{rR^2}{2}.$$

Hence, we can upper-bound the function suboptimality gap in problem (89) as follows:

$$\bar{f}(x) - \bar{f}(x^*) \leq p(x) - \min_{x' \in \mathbb{R}^d} p(x') + \frac{rR^2}{2}.$$

Let the regularization parameter $r > 0$ be chosen as follows:

$$r = \epsilon/R^2. \tag{90}$$

Then, we obtain the following:

$$\bar{f}(x) - \bar{f}(x^*) \leq p(x) - \min_{x' \in \mathbb{R}^d} p(x') + \frac{\epsilon}{2}. \tag{91}$$

We can apply Algorithm 1 to solving the regularized problem (1) with the regularization parameter $r$ defined in eq. (90). Theorem 3 implies that, to reach precision

$$p(x_o^K) - \min_{x' \in \mathbb{R}^d} p(x') \leq \frac{\epsilon}{2} \tag{92}$$

it is sufficient to perform the following number of decentralized communications:

$$K = \mathcal{O}\left(\frac{\chi M}{\sqrt{r\epsilon}}\right) \stackrel{(a)}{=} \mathcal{O}\left(\frac{\chi MR}{\epsilon}\right), \tag{93}$$

and the following number of subgradient computations:

$$K \times T = \mathcal{O}\left(\frac{M^2}{r\epsilon}\right) \stackrel{(b)}{=} \mathcal{O}\left(\frac{M^2R^2}{\epsilon^2}\right), \tag{94}$$

where (a) and (b) use the definition of $r$ in eq. (90). Using eqs. (91) and (92), we also obtain the desired precision $\bar{f}(x_o^K) - \bar{f}(x^*) \leq \epsilon$, which concludes the proof. $\qquad\square$

# E   Proofs of Lemmas from Section D.1

## E.1   Proof of Lemma 6

First, we pick the solution $x^* \in \mathbb{R}^d$ to problem (1), which is unique due to Assumption 3 and the fact that $r > 0$. Next, we define $w^* \in \mathcal{L}$ as follows:

$$w^* = (x^*, \ldots, x^*). \tag{95}$$

From Assumptions 1 and 2 it follows that $\operatorname{dom} p(x) = \mathbb{R}^d$ and $\operatorname{dom} f_i(x) = \mathbb{R}^d$ for all $i \in \{1, \ldots, n\}$, which implies the following:

$$0 \in \partial p(x^*) = rx^* + \frac{1}{n} \sum_{i=1}^n \partial f_i(x^*). \tag{96}$$

Hence, there exists a vector $\Delta^* = (\Delta_1^*, \ldots, \Delta_n^*) \in (\mathbb{R}^d)^n$ such that $\Delta_i^* \in \partial f_i(x^*)$ for all $i \in \{1, \ldots, n\}$, and the following relation holds:

$$rx^* + \frac{1}{n} \sum_{i=1}^n \Delta_i^* = 0. \tag{97}$$

Next, we define $y^* \in (\mathbb{R}^d)^n$ as follows:

$$y^* = \Delta^* + r_x w^*. \tag{98}$$

From Assumptions 1 and 2 it follows that $\operatorname{dom} F(x) = (\mathbb{R}^d)^n$, which implies $y^* \in \partial F(w^*)$ and $0 \in \partial(F(\cdot) - \langle y^*, \cdot \rangle)(w^*) = \partial_x Q(w^*, y^*, z^*)$.

Next, we define $z^* \in \mathcal{L}^\perp$ as follows:

$$z^* = -rw^* - \Delta^*. \tag{99}$$

Note that the inclusion $z^* \in \mathcal{L}^\perp$ is implied by eq. (97). Further, we get

$$\nabla_z Q(w^*, y^*, z^*) \overset{(a)}{=} -r_{yz}(y^* + z^*) \overset{(b)}{=} -r_{yz}(r_x - r)w^* \in \mathcal{L},$$

where (a) uses the definition of $Q(x, y, z)$ in eq. (14); (b) uses the definition of $y^*$ and $z^*$, and the last inclusion follows from the definition of $w^*$. Moreover, we obtain the following

$$\nabla_y Q(w^*, y^*, z^*) \overset{(a)}{=} -w^* - r_{yz}(y^* + z^*) \overset{(b)}{=} -r_{yz}(r_{yz}^{-1} + r_x - r)w^* \overset{(c)}{=} 0,$$

where (a) uses the definition of $Q(x, y, z)$ in eq. (14); (b) uses the definition of $y^*$ and $z^*$; (c) uses eq. (13).

From Assumption 2 it follows that $\|\Delta_i^*\| \leq M$ for all $i \in \{1, \ldots, n\}$. Hence, using eq. (97), we get $r\|x^*\| \leq M$, which implies $\|w^*\|^2 \leq nM^2/r^2$. Moreover, we get

$$\|y^*\| \leq \|\Delta^*\| + r_x \|w^*\| \leq \sqrt{n}(M + r_x M/r) = (1 + r_x/r)\sqrt{n}M,$$

which implies $\|y^*\|^2 \leq (1 + r_x/r)^2 nM^2$. Finally, we obtain

$$\|z^*\| \leq r\|w^*\| + \|\Delta^*\| \leq 2\sqrt{n}M,$$

which implies $\|z^*\|^2 \leq 4nM^2$ and concludes the proof. $\qquad \square$

## E.2  Proof of Lemma 7

We start with the following upper-bound on $\frac{1}{2\eta_x^k}\|x^{k,t+1}-x\|^2$:

$$\frac{1}{2\eta_x^k}\|x^{k,t+1}-x\|^2$$

$$\overset{(a)}{=} \frac{1}{2\eta_x^k}\|x^{k,t}-x\|^2 - \frac{1}{2\eta_x^k}\|x^{k,t+1}-x^{k,t}\|^2 + \frac{1}{\eta_x^k}\langle x^{k,t+1}-x^{k,t}, x^{k,t+1}-x\rangle$$

$$\overset{(b)}{=} \frac{1}{2\eta_x^k}\|x^{k,t}-x\|^2 - \frac{1}{2\eta_x^k}\|x^{k,t+1}-x^{k,t}\|^2$$
$$- \langle g_x^{k,t} + \beta_k x^{k,t+1} - y^{k+1} + \tau_x^k(x^{k,t+1}-x^k), x^{k,t+1}-x\rangle$$

$$= \frac{1}{2\eta_x^k}\|x^{k,t}-x\|^2 - \frac{1}{2\eta_x^k}\|x^{k,t+1}-x^{k,t}\|^2 + \langle y^{k+1}, x^{k,t+1}-x\rangle$$
$$- \langle \beta_k x^{k,t+1} + \tau_x^k(x^{k,t+1}-x^k), x^{k,t+1}-x\rangle - \langle g_x^{k,t}, x^{k,t+1}-x\rangle$$

$$\overset{(c)}{\leq} \frac{1}{2\eta_x^k}\|x^{k,t}-x\|^2 - \frac{1}{2\eta_x^k}\|x^{k,t+1}-x^{k,t}\|^2 + \langle y^{k+1}, x^{k,t+1}-x\rangle$$
$$- \frac{\tau_x^k}{2}\|x^{k,t+1}-x^k\|^2 - \frac{\tau_x^k}{2}\|x^{k,t+1}-x\|^2 + \frac{\tau_x^k}{2}\|x^k-x\|^2$$
$$- \frac{\beta_k}{2}\|x^{k,t+1}\|^2 - \frac{\beta_k}{2}\|x^{k,t+1}-x\|^2 + \frac{\beta_k}{2}\|x\|^2 - \langle g_x^{k,t}, x^{k,t}-x\rangle - \langle g_x^{k,t}, x^{k,t+1}-x^{k,t}\rangle$$

$$\overset{(d)}{\leq} \frac{1}{2\eta_x^k}\|x^{k,t}-x\|^2 - \frac{1}{2\eta_x^k}\|x^{k,t+1}-x^{k,t}\|^2 + \langle y^{k+1}, x^{k,t+1}-x\rangle$$
$$- \frac{\tau_x^k}{2}\|x^{k,t+1}-x^k\|^2 - \frac{\tau_x^k}{2}\|x^{k,t+1}-x\|^2 + \frac{\tau_x^k}{2}\|x^k-x\|^2$$
$$- \frac{\beta_k}{2}\|x^{k,t+1}\|^2 - \frac{\beta_k}{2}\|x^{k,t+1}-x\|^2 + \frac{\beta_k}{2}\|x\|^2$$
$$+ \sum_{i=1}^n (f_i(x_i) - f_i(x_i^{k,t}) - \langle g_{x,i}^{k,t}, x_i^{k,t+1}-x_i^{k,t}\rangle),$$

where $(g_{x,1}^{k,t},\ldots,g_{x,n}^{k,t}) = (\hat\nabla f_1(x_1^{k,t}),\ldots,\hat\nabla f_n(x_n^{k,t})) = g_x^{k,t} \in (\mathbb{R}^d)^n$, (a) and (c) uses the parallelogram rule; (b) uses line 12 of Algorithm 1; (d) uses line 11 of Algorithm 1, Definition 1 and Assumption 1. Further, we obtain

$$\frac{1}{2\eta_x^k}\|x^{k,t+1}-x\|^2$$

$$\overset{(a)}{\leq} \frac{1}{2\eta_x^k}\|x^{k,t}-x\|^2 - \frac{1}{2\eta_x^k}\|x^{k,t+1}-x^{k,t}\|^2 + \langle y^{k+1}, x^{k,t+1}-x\rangle$$
$$- \frac{\tau_x^k}{2}\|x^{k,t+1}-x^k\|^2 - \frac{\tau_x^k}{2}\|x^{k,t+1}-x\|^2 + \frac{\tau_x^k}{2}\|x^k-x\|^2$$
$$- \frac{\beta_k}{2}\|x^{k,t+1}\|^2 - \frac{\beta_k}{2}\|x^{k,t+1}-x\|^2 + \frac{\beta_k}{2}\|x\|^2$$
$$+ \sum_{i=1}^n (f_i(x_i) - f_i(x_i^{k,t+1}) + M\|x_i^{k,t+1}-x_i^{k,t}\| + \|g_{x,i}^{k,t}\|\|x_i^{k,t+1}-x_i^{k,t}\|)$$

$$\overset{(b)}{\leq} \frac{1}{2\eta_x^k}\|x^{k,t}-x\|^2 - \frac{1}{2\eta_x^k}\|x^{k,t+1}-x^{k,t}\|^2 + \langle y^{k+1}, x^{k,t+1}-x\rangle$$
$$- \frac{\tau_x^k}{2}\|x^{k,t+1}-x^k\|^2 - \frac{\tau_x^k}{2}\|x^{k,t+1}-x\|^2 + \frac{\tau_x^k}{2}\|x^k-x\|^2$$
$$- \frac{\beta_k}{2}\|x^{k,t+1}\|^2 - \frac{\beta_k}{2}\|x^{k,t+1}-x\|^2 + \frac{\beta_k}{2}\|x\|^2$$
$$+ \sum_{i=1}^n (f_i(x_i) - f_i(x_i^{k,t+1}) + 2M\|x_i^{k,t+1}-x_i^{k,t}\|)$$

$$\overset{(c)}{\leq} \frac{1}{2\eta_x^k}\|x^{k,t}-x\|^2 - \frac{1}{2\eta_x^k}\|x^{k,t+1}-x^{k,t}\|^2 + \langle y^{k+1}, x^{k,t+1}-x\rangle$$

$$- \frac{\tau_x^k}{2}\|x^{k,t+1}-x^k\|^2 - \frac{\tau_x^k}{2}\|x^{k,t+1}-x\|^2 + \frac{\tau_x^k}{2}\|x^k-x\|^2$$

$$+ F(x) - F(x^{k,t+1}) - \frac{r_x}{2}\|x^{k,t+1}-x\|^2 + \sum_{i=1}^n \left(\frac{1}{2\eta_x^k}\|x_i^{k,t+1}-x_i^{k,t}\|^2 + 2\eta_x^k M^2\right)$$

$$= \frac{1}{2\eta_x^k}\|x^{k,t}-x\|^2 + \frac{\tau_x^k}{2}\|x^k-x\|^2 + \langle y^{k+1}, x^{k,t+1}-x\rangle + 2n\eta_x^k M^2$$

$$- \frac{\tau_x^k}{2}\|x^{k,t+1}-x^k\|^2 - \frac{\tau_x^k+r_x}{2}\|x^{k,t+1}-x\|^2 - F(x^{k,t+1}) + F(x),$$

where (a) uses Assumption 2 and the Cauchy-Schwarz inequality; (b) uses the inequality $\|g_{x,i}^{k,t}\| \leq M$, which follows from Assumption 2; (c) uses the definition of $\beta_k$ in eq. (75), the definition of $F(x)$ in eq. (12) and Young's inequality. After rearranging, we obtain

$$\frac{1}{2\eta_x^k}\|x^{k,t+1}-x\|^2 \leq \frac{1}{2\eta_x^k}\|x^{k,t}-x\|^2 + \frac{\tau_x^k}{2}\|x^k-x\|^2 + 2n\eta_x^k M^2 - \Delta^{k,t+1},$$

where $\Delta^{k,t+1}$ is defined as

$$\Delta^{k,t+1} = F(x^{k,t+1}) - F(x) - \langle y^{k+1}, x^{k,t+1}-x\rangle$$

$$+ \frac{\tau_x^k+r_x}{2}\|x^{k,t+1}-x\|^2 + \frac{\tau_x^k}{2}\|x^{k,t+1}-x^k\|^2$$

Now, we sum these inequalities for $t = 0, \ldots, T-1$ and obtain

$$\frac{1}{2\eta_x^k}\|x^{k,T}-x\|^2 \leq \frac{1}{2\eta_x^k}\|x^{k,0}-x\|^2 + \frac{\tau_x^k T}{2}\|x^k-x\|^2 + 2n\eta_x^k M^2 T - \sum_{t=1}^T \Delta^{k,t}.$$

Dividing both sides of the inequality by $T$ gives

$$\frac{1}{2\eta_x^k T}\|x^{k,T}-x\|^2 \leq \frac{1}{2\eta_x^k T}\|x^{k,0}-x\|^2 + \frac{\tau_x^k}{2}\|x^k-x\|^2 + 2n\eta_x^k M^2 - \frac{1}{T}\sum_{t=1}^T \Delta^{k,t}.$$

Using the definition of $\Delta^{k,t}$, the definition of $\tilde{x}^k$ on line 13 of Algorithm 1 and Assumption 1, we obtain

$$\frac{1}{2\eta_x^k T}\|x^{k,T}-x\|^2 \leq \frac{1}{2\eta_x^k T}\|x^{k,0}-x\|^2 + \frac{\tau_x^k}{2}\|x^k-x\|^2 + 2n\eta_x^k M^2$$

$$- \left(F(\tilde{x}^{k+1}) - F(x) - \langle y^{k+1}, \tilde{x}^{k+1}-x\rangle\right)$$

$$- \left(\frac{\tau_x^k+r_x}{2}\|\tilde{x}^{k+1}-x\|^2 + \frac{\tau_x^k}{2}\|\tilde{x}^{k+1}-x^k\|^2\right).$$

Using the definition of $\eta_x^k$ and $\beta_k$ in eq. (75), we obtain

$$\frac{\tau_x^k}{2}\|x^{k,T}-x\|^2 + \frac{\tau_x^k+\beta_k}{2}\|\tilde{x}^{k+1}-x\|^2 \leq \frac{\tau_x^k}{2}\|x^{k,0}-x\|^2 + \frac{\tau_x^k}{2}\|x^k-x\|^2 + 2n\eta_x^k M^2$$

$$- \left(F(\tilde{x}^{k+1}) - F(x) - \langle y^{k+1}, \tilde{x}^{k+1}-x\rangle + \frac{\tau_x^k}{2}\|\tilde{x}^{k+1}-x^k\|^2\right).$$

Using the definition of $x^{k,0}$ on line 9 of Algorithm 1, the definition of $x^{k+1}$ on line 13 of Algorithm 1, the definition of $\eta_x^k$, $\beta_k$ and $\sigma_k$ in eq. (75) and the convexity of $\|\cdot\|$, we obtain

$$(\tau_x^k + \tfrac{1}{2}r_x)\|x^{k+1}-x\|^2 \leq \tau_x^k\|x^k-x\|^2 + \frac{2nM^2}{\tau_x^k T}$$

$$- \left(F(\tilde{x}^{k+1}) - F(x) - \langle y^{k+1}, \tilde{x}^{k+1}-x\rangle + \frac{\tau_x^k}{2}\|\tilde{x}^{k+1}-x^k\|^2\right),$$

which concludes the proof. $\qquad\square$

### E.3 Proof of Lemma 8

Using Assumption 4, and the definition of $\mathbf{P}$ in eq. (78), we obtain

$$\mathbf{P}(\mathbf{W}_k \otimes \mathbf{I}_d) = (\mathbf{W}_k \otimes \mathbf{I}_d)\mathbf{P} = (\mathbf{W}_k \otimes \mathbf{I}_d). \tag{100}$$

Then, the desired relations can be trivially obtained by analyzing the lines of Algorithm 1. $\qquad\square$

### E.4 Proof of Lemma 9

We can upper-bound $\|\eta_z^{k+1} m^{k+1}\|_{\mathbf{P}}^2$ as follows:

$$
\begin{aligned}
\|\eta_z^{k+1} m^{k+1}\|_{\mathbf{P}}^2 &\overset{(a)}{=} \|\eta_z^k(m^k + g_z^k - \hat{g}_z^k)\|_{\mathbf{P}}^2 \\
&\overset{(b)}{=} \|\eta_z^k(m^k + g_z^k - (\mathbf{W}_k \otimes \mathbf{I}_d)(m^k + g_z^k))\|_{\mathbf{P}}^2 \\
&\overset{(c)}{=} \|\eta_z^k(\mathbf{P}(m^k + g_z^k) - (\mathbf{W}_k \otimes \mathbf{I}_d)\mathbf{P}(m^k + g_z^k))\|^2 \\
&\overset{(d)}{\leq} (1 - 1/\chi)\|\eta_z^k \mathbf{P}(m^k + g_z^k)\|^2 \\
&\overset{(e)}{\leq} (1 - 1/\chi)\left((1 + 1/(2\chi))\|\eta_z^k m^k\|_{\mathbf{P}}^2 + (1 + 2\chi)\|\eta_z^k g_z^k\|_{\mathbf{P}}^2\right) \\
&\leq (1 - 1/(2\chi))\|\eta_z^k m^k\|_{\mathbf{P}}^2 + 2\chi\|\eta_z^k g_z^k\|_{\mathbf{P}}^2
\end{aligned}
$$

where (a) uses line 8 of Algorithm 1; (b) uses line 6 of Algorithm 1; (c) uses eq. (100); (d) uses Assumption 5; (e) uses the parallelogram rule and Young's inequality. Using this, we obtain

$$\|\eta_z^k m^k\|_{\mathbf{P}}^2 \leq 2\chi\|\eta_z^k m^k\|_{\mathbf{P}}^2 - 2\chi\|\eta_z^{k+1} m^{k+1}\|_{\mathbf{P}}^2 + 4\chi^2\|\eta_z^k g_z^k\|_{\mathbf{P}}^2,$$

which concludes the proof. $\qquad\square$

### E.5 Proof of Lemma 10

We can upper bound $\|\tilde{g}_z^k - \mathbf{P}g_z^k\|^2$ as follows:

$$
\begin{aligned}
\|\tilde{g}_z^k - \mathbf{P}g_z^k\|^2 &\overset{(a)}{=} \|(\mathbf{W}_k \otimes \mathbf{I}_d)g_z^k - \mathbf{P}g_z^k\|^2 \\
&\overset{(b)}{=} \|(\mathbf{W}_k \otimes \mathbf{I}_d)\mathbf{P}g_z^k - \mathbf{P}g_z^k\|^2 \\
&\overset{(c)}{\leq} (1 - 1/\chi)\|g_z^k\|_{\mathbf{P}}^2
\end{aligned}
$$

where (a) uses line 6 of Algorithm 1; (b) uses eq. (100); (c) uses Assumption 5. On the other hand, $\|\tilde{g}_z^k - \mathbf{P}g_z^k\|^2$ is equal to the following:

$$
\begin{aligned}
\|\tilde{g}_z^k - \mathbf{P}g_z^k\|^2 &\overset{(a)}{=} \|\tilde{g}_z^k\|^2 + \|g_z^k\|_{\mathbf{P}}^2 - 2\langle\tilde{g}_z^k, \mathbf{P}g_z^k\rangle \\
&\overset{(b)}{=} \frac{1}{(\theta_z^k)^2}\|\overline{z}^{k+1} - \underline{z}^k\|^2 + \|g_z^k\|_{\mathbf{P}}^2 + \frac{2}{\theta_z^k}\langle\overline{z}^{k+1} - \underline{z}^k, \mathbf{P}g_z^k\rangle.
\end{aligned}
$$

where (a) uses the parallelogram rule; (b) uses line 8 of Algorithm 1. Hence, we obtain the following

$$\frac{1}{(\theta_z^k)^2}\|\overline{z}^{k+1} - \underline{z}^k\|^2 + \frac{2}{\theta_z^k}\langle\overline{z}^{k+1} - \underline{z}^k, \mathbf{P}g_z^k\rangle + \frac{1}{\chi}\|g_z^k\|_{\mathbf{P}}^2 \leq 0.$$

After rearranging and multiplying both sides of the inequality by $\frac{\theta_z^k}{2\alpha_k}$, we obtain

$$
\begin{aligned}
0 &\geq \alpha_k^{-1}\left(\langle\overline{z}^{k+1} - \underline{z}^k, \mathbf{P}g_z^k\rangle + \frac{1}{2\theta_z^k}\|\overline{z}^{k+1} - \underline{z}^k\|^2\right) + \frac{\theta_z^k}{2\alpha_k\chi}\|g_z^k\|_{\mathbf{P}}^2 \\
&\overset{(a)}{=} \alpha_k^{-1}\left(\langle\mathbf{P}(\overline{z}^{k+1} - \underline{z}^k), g_z^k\rangle + r_{yz}\|\overline{z}^{k+1} - \underline{z}^k\|^2\right) + \frac{1}{4\alpha_k\chi r_{yz}}\|g_z^k\|_{\mathbf{P}}^2 \\
&\overset{(b)}{=} \alpha_k^{-1}\left(\langle\overline{z}^{k+1} - \underline{z}^k, g_z^k\rangle + r_{yz}\|\overline{z}^{k+1} - \underline{z}^k\|^2\right) + \frac{1}{4\alpha_k\chi r_{yz}}\|g_z^k\|_{\mathbf{P}}^2
\end{aligned}
$$

where (a) uses eq. (80); (b) uses Lemma 8, which concludes the proof. $\qquad\square$

## E.6 Proof of Lemma 11

We can upper-bound $\frac{1}{2\eta_y^k}\|y^{k+1}-y\|^2$ as follows:

$$
\frac{1}{2\eta_y^k}\|y^{k+1}-y\|^2 \overset{(a)}{=} \frac{1}{2\eta_y^k}\|y^k-y\|^2 - \frac{1}{2\eta_y^k}\|y^{k+1}-y^k\|^2 + \frac{1}{\eta_y^k}\langle y^{k+1}-y^k, y^{k+1}-y\rangle
$$

$$
\overset{(b)}{=} \frac{1}{2\eta_y^k}\|y^k-y\|^2 - \frac{1}{2\eta_y^k}\|y^{k+1}-y^k\|^2 - \langle g_y^k + \hat{x}^{k+1}, y^{k+1}-y\rangle
$$

$$
= \frac{1}{2\eta_y^k}\|y^k-y\|^2 - \frac{1}{2\eta_y^k}\|y^{k+1}-y^k\|^2 - \langle \hat{x}^{k+1}, y^{k+1}-y\rangle
$$

$$
- \langle g_y^k, y^{k+1}-y^k + y^k - \underline{y}^k + \underline{y}^k - y\rangle
$$

$$
\overset{(c)}{=} \frac{1}{2\eta_y^k}\|y^k-y\|^2 - \frac{1}{2\eta_y^k}\|y^{k+1}-y^k\|^2 - \langle \hat{x}^{k+1}, y^{k+1}-y\rangle
$$

$$
- \alpha_k^{-1}\langle g_y^k, \overline{y}^{k+1}-\underline{y}^k\rangle + (1-\alpha_k)\alpha_k^{-1}\langle g_y^k, \overline{y}^k - \underline{y}^k\rangle + \langle g_y^k, y - \underline{y}^k\rangle
$$

$$
= \frac{1}{2\eta_y^k}\|y^k-y\|^2 - \frac{1}{2\eta_y^k}\|y^{k+1}-y^k\|^2 - \langle \tilde{x}^{k+1}, y^{k+1}-y\rangle
$$

$$
- \alpha_k^{-1}\langle g_y^k, \overline{y}^{k+1}-\underline{y}^k\rangle + (1-\alpha_k)\alpha_k^{-1}\langle g_y^k, \overline{y}^k - \underline{y}^k\rangle + \langle g_y^k, y - \underline{y}^k\rangle
$$

$$
+ \langle \tilde{x}^{k+1} - \hat{x}^{k+1}, y^{k+1}-y\rangle
$$

where (a) uses the parallelogram rule; (b) uses line 7 of Algorithm 1; (c) uses Lines 4 and 8 of Algorithm 1. Further, we can upper-bound the term $\langle \tilde{x}^{k+1} - \hat{x}^{k+1}, y^{k+1}-y\rangle$ as follows:

$$
\langle \tilde{x}^{k+1} - \hat{x}^{k+1}, y^{k+1}-y\rangle
$$

$$
\overset{(a)}{=} \langle \tilde{x}^{k+1} - x^k - \gamma_k(\tilde{x}^k - x^{k-1}), y^{k+1}-y\rangle
$$

$$
= \gamma_k\langle x^{k-1}-\tilde{x}^k, y^k - y\rangle - \langle x^k - \tilde{x}^{k+1}, y^{k+1}-y\rangle + \gamma_k\langle x^{k-1}-\tilde{x}^k, y^{k+1}-y^k\rangle
$$

$$
\overset{(b)}{\leq} \gamma_k\langle x^{k-1}-\tilde{x}^k, y^k - y\rangle - \langle x^k - \tilde{x}^{k+1}, y^{k+1}-y\rangle + \frac{1}{4\eta_y^k}\|y^{k+1}-y^k\|^2
$$

$$
+ 2\eta_y^k\gamma_k^2\|x^{k-1}-\tilde{x}^k\|^2.
$$

where (a) uses Line 7 of Algorithm 1; (b) uses Young's inequality. Plugging this into the previous inequality gives

$$
\frac{1}{2\eta_y^k}\|y^{k+1}-y\|^2 \leq \frac{1}{2\eta_y^k}\|y^k-y\|^2 - \frac{1}{4\eta_y^k}\|y^{k+1}-y^k\|^2 - \langle \tilde{x}^{k+1}, y^{k+1}-y\rangle
$$

$$
- \alpha_k^{-1}\langle g_y^k, \overline{y}^{k+1}-\underline{y}^k\rangle + (1-\alpha_k)\alpha_k^{-1}\langle g_y^k, \overline{y}^k - \underline{y}^k\rangle + \langle g_y^k, y - \underline{y}^k\rangle
$$

$$
+ \gamma_k\langle x^{k-1}-\tilde{x}^k, y^k - y\rangle - \langle x^k - \tilde{x}^{k+1}, y^{k+1}-y\rangle + 2\eta_y^k\gamma_k^2\|x^{k-1}-\tilde{x}^k\|^2
$$

$$
\overset{(a)}{=} \frac{1}{2\eta_y^k}\|y^k-y\|^2 - \langle \tilde{x}^{k+1}, y^{k+1}-y\rangle + 2\eta_y^k\gamma_k^2\|x^{k-1}-\tilde{x}^k\|^2
$$

$$
+ \langle g_y^k, y - \underline{y}^k\rangle + (1-\alpha_k)\alpha_k^{-1}\langle g_y^k, \overline{y}^k - \underline{y}^k\rangle - \alpha_k^{-1}\langle g_y^k, \overline{y}^{k+1}-\underline{y}^k\rangle
$$

$$
- \frac{1}{4\eta_y^k\alpha_k^2}\|\overline{y}^{k+1}-\underline{y}^k\|^2 + \gamma_k\langle x^{k-1}-\tilde{x}^k, y^k - y\rangle - \langle x^k - \tilde{x}^{k+1}, y^{k+1}-y\rangle
$$

$$
\overset{(b)}{=} \frac{1}{2\eta_y^k}\|y^k-y\|^2 + 2\eta_y^k\gamma_k^2\|x^{k-1}-\tilde{x}^k\|^2 - \langle \tilde{x}^{k+1}, y^{k+1}-y\rangle
$$

$$
+ \gamma_k\langle x^{k-1}-\tilde{x}^k, y^k - y\rangle - \langle x^k - \tilde{x}^{k+1}, y^{k+1}-y\rangle + \langle g_y^k, y - \underline{y}^k\rangle
$$

$$
+ (1-\alpha_k)\alpha_k^{-1}\langle g_y^k, \overline{y}^k - \underline{y}^k\rangle - \alpha_k^{-1}\left(\langle g_y^k, \overline{y}^{k+1}-\underline{y}^k\rangle + r_{yz}\|\overline{y}^{k+1}-\underline{y}^k\|^2\right),
$$

where (a) line 8 of Algorithm 1; (b) uses eqs. (83) and (84).

Let $\hat{z}^k$ be defined for all $k \in \{0, \dots, K\}$ as follows:

$$
\hat{z}^k = z^k - \eta_z^k \mathbf{P}m^k. \tag{101}
$$

Using eq. (101) and lines 7 and 8 of Algorithm 1, we obtain

$$
\begin{aligned}
\hat{z}^{k+1} &= \hat{z}^k + z^{k+1} - z^k - \mathbf{P}(\eta_z^{k+1} m^{k+1} - \eta_z^k m^k) \\
&= \hat{z}^k - \mathbf{P}(\eta_z^k \hat{g}_z^k + \eta_z^{k+1} m^{k+1} - \eta_z^k m^k) \\
&= \hat{z}^k - \mathbf{P}(\eta_z^k \hat{g}_z^k + \eta_z^{k+1}(\eta_z^k/\eta_z^{k+1})(m^k + g_z^k - \hat{g}_z^k) - \eta_z^k m^k) \\
&= \hat{z}^k - \eta_z^k \mathbf{P} g_z^k.
\end{aligned}
$$

Hence, we can upper-bound $\frac{1}{2\eta_z^k}\|\hat{z}^{k+1} - z\|^2$ as follows:

$$
\begin{aligned}
\frac{1}{2\eta_z^k}\|\hat{z}^{k+1} - z\|^2 &\overset{(a)}{=} \frac{1}{2\eta_z^k}\|\hat{z}^k - z\|^2 + \frac{1}{2\eta_z^k}\|\hat{z}^{k+1} - \hat{z}^k\|^2 + \frac{1}{\eta_z^k}\langle \hat{z}^{k+1} - \hat{z}^k, \hat{z}^k - z\rangle \\
&\overset{(b)}{=} \frac{1}{2\eta_z^k}\|\hat{z}^k - z\|^2 + \frac{\eta_z^k}{2}\|g_z^k\|_{\mathbf{P}}^2 - \langle \mathbf{P}g_z^k, \hat{z}^k - z\rangle \\
&= \frac{1}{2\eta_z^k}\|\hat{z}^k - z\|^2 + \frac{\eta_z^k}{2}\|g_z^k\|_{\mathbf{P}}^2 - \langle \mathbf{P}g_z^k, z^k - z\rangle + \langle \mathbf{P}g_z^k, z^k - \hat{z}^k\rangle \\
&\overset{(c)}{=} \frac{1}{2\eta_z^k}\|\hat{z}^k - z\|^2 + \frac{\eta_z^k}{2}\|g_z^k\|_{\mathbf{P}}^2 - \langle \mathbf{P}g_z^k, z^k - z\rangle + \eta_z^k \langle \mathbf{P}g_z^k, \mathbf{P}m^k\rangle,
\end{aligned}
$$

where (a) uses the parallelogram rule; (b) uses the update rule for $\hat{z}^k$ which we previously obtained; (c) uses eq. (101). Further, we can upper-bound the term $\eta_z^k \langle \mathbf{P}g_z^k, \mathbf{P}m^k\rangle$ as follows

$$
\begin{aligned}
\eta_z^k \langle \mathbf{P}g_z^k, \mathbf{P}m^k\rangle &\overset{(a)}{\leq} \frac{1}{\eta_z^k}\|\eta_z^k g_z^k\|_{\mathbf{P}}\|\eta_z^k m^k\|_{\mathbf{P}} \\
&\overset{(b)}{\leq} \frac{1}{2\eta_z^k}\left(2\chi\|\eta_z^k g_z^k\|_{\mathbf{P}}^2 + \frac{1}{2\chi}\|\eta_z^k m^k\|_{\mathbf{P}}^2\right) \\
&\overset{(c)}{\leq} \frac{1}{2\eta_z^k}\left(4\chi\|\eta_z^k g_z^k\|_{\mathbf{P}}^2 + \|\eta_z^k m^k\|_{\mathbf{P}}^2 - \|\eta_z^{k+1} m^{k+1}\|_{\mathbf{P}}^2\right)
\end{aligned}
$$

where (a) uses the Cauchy-Schwarz inequality; (b) uses Young's inequality; (c) uses Lemma 9. Plugging this into the previous inequality gives

$$
\begin{aligned}
\frac{1}{2\eta_z^k}\|\hat{z}^{k+1} - z\|^2 &\leq \frac{1}{2\eta_z^k}\|\hat{z}^k - z\|^2 + \frac{\eta_z^k}{2}\|g_z^k\|_{\mathbf{P}}^2 - \langle \mathbf{P}g_z^k, z^k - z\rangle \\
&\quad + \frac{1}{2\eta_z^k}\left(4\chi\|\eta_z^k g_z^k\|_{\mathbf{P}}^2 + \|\eta_z^k m^k\|_{\mathbf{P}}^2 - \|\eta_z^{k+1} m^{k+1}\|_{\mathbf{P}}^2\right) \\
&\overset{(a)}{=} \frac{1}{2\eta_z^k}\|\hat{z}^k - z\|^2 + \frac{\eta_z^k(1 + 4\chi)}{2}\|g_z^k\|_{\mathbf{P}}^2 - \langle g_z^k, z^k - \underline{z}^k + \underline{z}^k - z\rangle \\
&\quad + \frac{1}{2\eta_z^k}\left(\|\eta_z^k m^k\|_{\mathbf{P}}^2 - \|\eta_z^{k+1} m^{k+1}\|_{\mathbf{P}}^2\right) \\
&\overset{(b)}{=} \frac{1}{2\eta_z^k}\|\hat{z}^k - z\|^2 + \frac{\eta_z^k(1 + 4\chi)}{2}\|g_z^k\|_{\mathbf{P}}^2 + \frac{1}{2\eta_z^k}\left(\|\eta_z^k m^k\|_{\mathbf{P}}^2 - \|\eta_z^{k+1} m^{k+1}\|_{\mathbf{P}}^2\right) \\
&\quad + \langle g_z^k, z - \underline{z}^k\rangle + (1 - \alpha_k)\alpha_k^{-1}\langle g_z^k, \overline{z}^k - \underline{z}^k\rangle \\
&\overset{(c)}{\leq} \frac{1}{2\eta_z^k}\|\hat{z}^k - z\|^2 + \frac{\eta_z^k(1 + 4\chi)}{2}\|g_z^k\|_{\mathbf{P}}^2 + \frac{1}{2\eta_z^k}\left(\|\eta_z^k m^k\|_{\mathbf{P}}^2 - \|\eta_z^{k+1} m^{k+1}\|_{\mathbf{P}}^2\right) \\
&\quad + \langle g_z^k, z - \underline{z}^k\rangle + (1 - \alpha_k)\alpha_k^{-1}\langle g_z^k, \overline{z}^k - \underline{z}^k\rangle \\
&\quad - \alpha_k^{-1}\left(\langle \overline{z}^{k+1} - \underline{z}^k, g_z^k\rangle + r_{yz}\|\overline{z}^{k+1} - \underline{z}^k\|^2\right) - \frac{1}{4\alpha_k \chi r_{yz}}\|g_z^k\|_{\mathbf{P}}^2 \\
&\overset{(d)}{\leq} \frac{1}{2\eta_z^k}\|\hat{z}^k - z\|^2 + \frac{1}{2\eta_z^k}\left(\|\eta_z^k m^k\|_{\mathbf{P}}^2 - \|\eta_z^{k+1} m^{k+1}\|_{\mathbf{P}}^2\right) + \langle g_z^k, z - \underline{z}^k\rangle \\
&\quad + (1 - \alpha_k)\alpha_k^{-1}\langle g_z^k, \overline{z}^k - \underline{z}^k\rangle - \alpha_k^{-1}\left(\langle \overline{z}^{k+1} - \underline{z}^k, g_z^k\rangle + r_{yz}\|\overline{z}^{k+1} - \underline{z}^k\|^2\right)
\end{aligned}
$$

where (a) uses Lemma 8 and the fact that $z \in \mathcal{L}^\perp$; (b) uses line 4 of Algorithm 1; (c) uses Lemma 10; (d) uses eqs. (83) and (84).

Now we combine the upper-bounds for $\frac{1}{2\eta_y^k}\|y^{k+1} - y\|^2$ and $\frac{1}{2\eta_z^k}\|\hat{z}^{k+1} - z\|^2$ and obtain the following:

$$\frac{1}{2\eta_y^k}\|y^{k+1} - y\|^2 + \frac{1}{2\eta_z^k}\|\hat{z}^{k+1} - z\|^2$$

$$\leq \frac{1}{2\eta_y^k}\|y^k - y\|^2 + \frac{1}{2\eta_z^k}\|\hat{z}^k - z\|^2 + 2\eta_y^k\gamma_k^2\|x^{k-1} - \tilde{x}^k\|^2 - \langle \tilde{x}^{k+1}, y^{k+1} - y\rangle$$

$$+ \gamma_k\langle x^{k-1} - \tilde{x}^k, y^k - y\rangle - \langle x^k - \tilde{x}^{k+1}, y^{k+1} - y\rangle + \frac{1}{2\eta_z^k}\left(\|\eta_z^k m^k\|_\mathbf{P}^2 - \|\eta_z^{k+1}m^{k+1}\|_\mathbf{P}^2\right)$$

$$+ \langle g_y^k, y - \underline{y}^k\rangle + \langle g_z^k, z - \underline{z}^k\rangle + (1 - \alpha_k)\alpha_k^{-1}\left(\langle g_y^k, \overline{y}^k - \underline{y}^k\rangle + \langle g_z^k, \overline{z}^k - \underline{z}^k\rangle\right)$$

$$- \alpha_k^{-1}\left(\langle g_y^k, \overline{y}^{k+1} - \underline{y}^k\rangle + \langle \overline{z}^{k+1} - \underline{z}^k, g_z^k\rangle + r_{yz}\|\overline{y}^{k+1} - \underline{y}^k\|^2 + r_{yz}\|\overline{z}^{k+1} - \underline{z}^k\|^2\right)$$

$$\overset{(a)}{\leq} \frac{1}{2\eta_y^k}\|y^k - y\|^2 + \frac{1}{2\eta_z^k}\|\hat{z}^k - z\|^2 + 2\eta_y^k\gamma_k^2\|x^{k-1} - \tilde{x}^k\|^2 - \langle \tilde{x}^{k+1}, y^{k+1} - y\rangle$$

$$+ \gamma_k\langle x^{k-1} - \tilde{x}^k, y^k - y\rangle - \langle x^k - \tilde{x}^{k+1}, y^{k+1} - y\rangle + \frac{1}{2\eta_z^k}\left(\|\eta_z^k m^k\|_\mathbf{P}^2 - \|\eta_z^{k+1}m^{k+1}\|_\mathbf{P}^2\right)$$

$$+ G(y, z) - G(\underline{y}^k, \underline{z}^k) + (1 - \alpha_k)\alpha_k^{-1}\left(G(\overline{y}^k, \overline{z}^k) - G(\underline{y}^k, \underline{z}^k)\right)$$

$$- \alpha_k^{-1}\left(G(\overline{y}^{k+1}, \overline{z}^{k+1}) - G(\underline{y}^k, \underline{z}^k)\right)$$

$$= \frac{1}{2\eta_y^k}\|y^k - y\|^2 + \frac{1}{2\eta_z^k}\|\hat{z}^k - z\|^2 + 2\eta_y^k\gamma_k^2\|x^{k-1} - \tilde{x}^k\|^2 - \langle \tilde{x}^{k+1}, y^{k+1} - y\rangle$$

$$+ \gamma_k\langle x^{k-1} - \tilde{x}^k, y^k - y\rangle - \langle x^k - \tilde{x}^{k+1}, y^{k+1} - y\rangle + \frac{1}{2\eta_z^k}\left(\|\eta_z^k m^k\|_\mathbf{P}^2 - \|\eta_z^{k+1}m^{k+1}\|_\mathbf{P}^2\right)$$

$$+ (1 - \alpha_k)\alpha_k^{-1}\left(G(\overline{y}^k, \overline{z}^k) - G(y, z)\right) - \alpha_k^{-1}\left(G(\overline{y}^{k+1}, \overline{z}^{k+1}) - G(y, z)\right),$$

where (a) uses the definition of $g_y^k$ and $g_z^k$ on line 5 of Algorithm 1 and the convexity and $(2r_{yz})$-smoothness of the function $G(y, z)$. Further, we divide both sides of the inequality by $\alpha_k$ and, using eq. (83), obtain the following:

$$\frac{1}{2\eta_y}\|y^{k+1} - y\|^2 + \frac{1}{2\eta_z}\|\hat{z}^{k+1} - z\|^2 + \frac{1}{2\eta_z}\|\eta_z^{k+1}m^{k+1}\|_\mathbf{P}^2$$

$$\leq \frac{1}{2\eta_y}\|y^k - y\|^2 + \frac{1}{2\eta_z}\|\hat{z}^k - z\|^2 + \frac{1}{2\eta_z}\|\eta_z^k m^k\|_\mathbf{P}^2 + 2\eta_y\alpha_k^{-2}\gamma_k^2\|x^{k-1} - \tilde{x}^k\|^2$$

$$+ \gamma_k\alpha_k^{-1}\langle x^{k-1} - \tilde{x}^k, y^k - y\rangle - \alpha_k^{-1}\langle x^k - \tilde{x}^{k+1}, y^{k+1} - y\rangle - \alpha_k^{-1}\langle \tilde{x}^{k+1}, y^{k+1} - y\rangle$$

$$+ (\alpha_k^{-2} - \alpha_k^{-1})\left(G(\overline{y}^k, \overline{z}^k) - G(y, z)\right) - \alpha_k^{-2}\left(G(\overline{y}^{k+1}, \overline{z}^{k+1}) - G(y, z)\right)$$

Next, we divide the inequality in Lemma 7 by $\alpha_k$ and, using the definition of $\tau_x^k$ and $\tau_x$ in eqs. (83) and (84), obtain the following:

$$\tau_x(\alpha_k^{-2} + \alpha_k^{-1})\|x^{k+1} - x\|^2 \leq \tau_x\alpha_k^{-2}\|x^k - x\|^2 - \frac{\tau_x\alpha_k^{-2}}{2}\|\tilde{x}^{k+1} - x^k\|^2 + \frac{2nM^2}{\tau_x T}$$

$$- \alpha_k^{-1}\left(F(\tilde{x}^{k+1}) - F(x) - \langle y^{k+1}, \tilde{x}^{k+1} - x\rangle\right).$$

Combining this inequality with the previous upper-bound gives the following:

$$\tau_x(\alpha_k^{-2} + \alpha_k^{-1})\|x^{k+1} - x\|^2 + \frac{1}{2\eta_y}\|y^{k+1} - y\|^2 + \frac{1}{2\eta_z}\|\hat{z}^{k+1} - z\|^2 + \frac{1}{2\eta_z}\|\eta_z^{k+1}m^{k+1}\|_\mathbf{P}^2$$

$$\leq \tau_x\alpha_k^{-2}\|x^k - x\|^2 + \frac{1}{2\eta_y}\|y^k - y\|^2 + \frac{1}{2\eta_z}\|\hat{z}^k - z\|^2 + \frac{1}{2\eta_z}\|\eta_z^k m^k\|_\mathbf{P}^2 + \frac{2nM^2}{\tau_x T}$$

$$- \frac{\tau_x\alpha_k^{-2}}{2}\|\tilde{x}^{k+1} - x^k\|^2 + 2\eta_y\alpha_k^{-2}\gamma_k^2\|x^{k-1} - \tilde{x}^k\|^2 - \alpha_k^{-1}\langle x^k - \tilde{x}^{k+1}, y^{k+1} - y\rangle$$

$$+ \gamma_k\alpha_k^{-1}\langle x^{k-1} - \tilde{x}^k, y^k - y\rangle - \alpha_k^{-1}\left(F(\tilde{x}^{k+1}) - F(x) + \langle y^{k+1}, x\rangle - \langle \tilde{x}^{k+1}, y\rangle\right)$$

$$+ (\alpha_k^{-2} - \alpha_k^{-1})\left(G(\overline{y}^k, \overline{z}^k) - G(y,z)\right) - \alpha_k^{-2}\left(G(\overline{y}^{k+1}, \overline{z}^{k+1}) - G(y,z)\right)$$

$$\overset{(a)}{=} \tau_x \alpha_k^{-2}\|x^k - x\|^2 + \frac{1}{2\eta_y}\|y^k - y\|^2 + \frac{1}{2\eta_z}\|\hat{z}^k - z\|^2 + \frac{1}{2\eta_z}\|\eta_z^k m^k\|_{\mathbf{P}}^2 + \frac{2nM^2}{\tau_x T}$$

$$- \frac{\tau_x \alpha_k^{-2}}{2}\|\tilde{x}^{k+1} - x^k\|^2 + 2\eta_y \alpha_k^{-2}\gamma_k^2\|x^{k-1} - \tilde{x}^k\|^2 - \alpha_k^{-1}\langle x^k - \tilde{x}^{k+1}, y^{k+1} - y\rangle$$

$$+ \gamma_k \alpha_k^{-1}\langle x^{k-1} - \tilde{x}^k, y^k - y\rangle - \alpha_k^{-1}\left(F(\tilde{x}^{k+1}) - \langle \tilde{x}^{k+1}, y\rangle - G(y,z)\right)$$

$$+ (\alpha_k^{-2} - \alpha_k^{-1})\left(G(\overline{y}^k, \overline{z}^k) + \langle \overline{y}^k, x\rangle - F(x)\right) - \alpha_k^{-2}\left(G(\overline{y}^{k+1}, \overline{z}^{k+1}) + \langle \overline{y}^{k+1}, x\rangle - F(x)\right)$$

$$\overset{(b)}{=} \tau_x \alpha_k^{-2}\|x^k - x\|^2 + \frac{1}{2\eta_y}\|y^k - y\|^2 + \frac{1}{2\eta_z}\|\hat{z}^k - z\|^2 + \frac{1}{2\eta_z}\|\eta_z^k m^k\|_{\mathbf{P}}^2 + \frac{2nM^2}{\tau_x T}$$

$$- \frac{\tau_x \alpha_k^{-2}}{2}\|\tilde{x}^{k+1} - x^k\|^2 + 2\eta_y \alpha_k^{-2}\gamma_k^2\|x^{k-1} - \tilde{x}^k\|^2 - \alpha_k^{-1}\langle x^k - \tilde{x}^{k+1}, y^{k+1} - y\rangle$$

$$+ \gamma_k \alpha_k^{-1}\langle x^{k-1} - \tilde{x}^k, y^k - y\rangle - (\alpha_k^{-2} - \alpha_k^{-1})Q(x, \overline{y}^k, \overline{z}^k) + \alpha_k^{-2}Q(x, \overline{y}^{k+1}, \overline{z}^{k+1})$$

$$- \alpha_k^{-1}Q(\tilde{x}^{k+1}, y, z)$$

$$\overset{(c)}{\leq} \tau_x \alpha_k^{-2}\|x^k - x\|^2 + \frac{1}{2\eta_y}\|y^k - y\|^2 + \frac{1}{2\eta_z}\|\hat{z}^k - z\|^2 + \frac{1}{2\eta_z}\|\eta_z^k m^k\|_{\mathbf{P}}^2 + \frac{2nM^2}{\tau_x T}$$

$$- \frac{\tau_x \alpha_k^{-2}}{2}\|\tilde{x}^{k+1} - x^k\|^2 + 2\eta_y \alpha_k^{-2}\gamma_k^2\|x^{k-1} - \tilde{x}^k\|^2 - \alpha_k^{-1}\langle x^k - \tilde{x}^{k+1}, y^{k+1} - y\rangle$$

$$+ \gamma_k \alpha_k^{-1}\langle x^{k-1} - \tilde{x}^k, y^k - y\rangle - (\alpha_k^{-2} - \alpha_k^{-1})Q(x, \overline{y}^k, \overline{z}^k) + \alpha_k^{-2}Q(x, \overline{y}^{k+1}, \overline{z}^{k+1})$$

$$- \alpha_k^{-2}Q(\overline{x}^{k+1}, y, z) + (\alpha_k^{-2} - \alpha_k^{-1})Q(\overline{x}^k, y, z)$$

$$= \tau_x \alpha_k^{-2}\|x^k - x\|^2 + \frac{1}{2\eta_y}\|y^k - y\|^2 + \frac{1}{2\eta_z}\|\hat{z}^k - z\|^2 + \frac{1}{2\eta_z}\|\eta_z^k m^k\|_{\mathbf{P}}^2 + \frac{2nM^2}{\tau_x T}$$

$$- \frac{\tau_x \alpha_k^{-2}}{2}\|\tilde{x}^{k+1} - x^k\|^2 + 2\eta_y \alpha_k^{-2}\gamma_k^2\|x^{k-1} - \tilde{x}^k\|^2 - \alpha_k^{-1}\langle x^k - \tilde{x}^{k+1}, y^{k+1} - y\rangle$$

$$+ \gamma_k \alpha_k^{-1}\langle x^{k-1} - \tilde{x}^k, y^k - y\rangle + (\alpha_k^{-2} - \alpha_k^{-1})\left(Q(\overline{x}^k, y, z) - Q(x, \overline{y}^k, \overline{z}^k)\right)$$

$$- \alpha_k^{-2}\left(Q(\overline{x}^{k+1}, y, z) - Q(x, \overline{y}^{k+1}, \overline{z}^{k+1})\right)$$

where (a) uses the fact that $y^{k+1} = \alpha_k^{-1}\overline{y}^{k+1} - (1 - \alpha_k)\alpha_k^{-1}\overline{y}^k$, which follows from lines 4 and 8 of Algorithm 1; (b) uses the definition of $Q(x,y,z)$ in eq. (14); (c) uses line 13 of Algorithm 1 and Assumption 1.

Further, let $\alpha_K = 3/(K+3)$. Then from eq. (82) it follows that $\alpha_k^{-2} + \alpha_k^{-1} \geq \alpha_{k+1}^{-2}$, $\gamma_k \alpha_k^{-1} = (k+2)/3$ and $\alpha_k^{-1} = (k+3)/3$. Hence, we obtain the following:

$$\tau_x \alpha_{k+1}^{-2}\|x^{k+1} - x\|^2 + \frac{1}{2\eta_y}\|y^{k+1} - y\|^2 + \frac{1}{2\eta_z}\|\hat{z}^{k+1} - z\|^2 + \frac{1}{2\eta_z}\|\eta_z^{k+1} m^{k+1}\|_{\mathbf{P}}^2$$

$$\leq \tau_x \alpha_k^{-2}\|x^k - x\|^2 + \frac{1}{2\eta_y}\|y^k - y\|^2 + \frac{1}{2\eta_z}\|\hat{z}^k - z\|^2 + \frac{1}{2\eta_z}\|\eta_z^k m^k\|_{\mathbf{P}}^2 + \frac{2nM^2}{\tau_x T}$$

$$- \frac{\tau_x(k+3)^2}{18}\|\tilde{x}^{k+1} - x^k\|^2 + \frac{4\eta_y(k+2)^2}{18}\|x^{k-1} - \tilde{x}^k\|^2 - \frac{k+3}{3}\langle x^k - \tilde{x}^{k+1}, y^{k+1} - y\rangle$$

$$+ \frac{k+2}{3}\langle x^{k-1} - \tilde{x}^k, y^k - y\rangle + (\alpha_k^{-2} - \alpha_k^{-1})\left(Q(\overline{x}^k, y, z) - Q(x, \overline{y}^k, \overline{z}^k)\right)$$

$$- \alpha_k^{-2}\left(Q(\overline{x}^{k+1}, y, z) - Q(x, \overline{y}^{k+1}, \overline{z}^{k+1})\right)$$

$$\overset{(a)}{=} \tau_x \alpha_k^{-2}\|x^k - x\|^2 + \frac{1}{2\eta_y}\|y^k - y\|^2 + \frac{1}{2\eta_z}\|\hat{z}^k - z\|^2 + \frac{1}{2\eta_z}\|\eta_z^k m^k\|_{\mathbf{P}}^2 + \frac{2nM^2}{\tau_x T}$$

$$- \frac{2\eta_y(k+3)^2}{9}\|\tilde{x}^{k+1} - x^k\|^2 + \frac{2\eta_y(k+2)^2}{9}\|x^{k-1} - \tilde{x}^k\|^2 - \frac{k+3}{3}\langle x^k - \tilde{x}^{k+1}, y^{k+1} - y\rangle$$

$$+ \frac{k+2}{3}\langle x^{k-1} - \tilde{x}^k, y^k - y\rangle + (\alpha_k^{-2} - \alpha_k^{-1})\left(Q(\overline{x}^k, y, z) - Q(x, \overline{y}^k, \overline{z}^k)\right)$$

$$- \alpha_k^{-2}\left(Q(\overline{x}^{k+1}, y, z) - Q(x, \overline{y}^{k+1}, \overline{z}^{k+1})\right),$$

where (a) uses eq. (84).

Next, we sum these inequalities for $k = 0, \ldots, K - 1$ and obtain the following:

$$\tau_x \alpha_K^{-2} \|x^K - x\|^2 + \frac{1}{2\eta_y}\|y^K - y\|^2 + \frac{1}{2\eta_z}\|\hat{z}^K - z\|^2 + \frac{1}{2\eta_z}\|\eta_z^K m^K\|_{\mathbf{P}}^2$$

$$\leq \tau_x \alpha_0^{-2}\|x^0 - x\|^2 + \frac{1}{2\eta_y}\|y^0 - y\|^2 + \frac{1}{2\eta_z}\|\hat{z}^0 - z\|^2 + \frac{1}{2\eta_z}\|\eta_z^0 m^0\|_{\mathbf{P}}^2 + \frac{2nM^2 K}{\tau_x T}$$

$$+ \frac{8\eta_y}{9}\|\tilde{x}^0 - x^{-1}\|^2 + \tfrac{2}{3}\langle x^{-1} - \tilde{x}^0, y^0 - y\rangle$$

$$- \frac{2\eta_y(K+2)^2}{9}\|\tilde{x}^K - x^{K-1}\|^2 - \tfrac{1}{3}(K+2)\langle x^{K-1} - \tilde{x}^K, y^K - y\rangle$$

$$+ \sum_{k=0}^{K-1}(\alpha_k^{-2} - \alpha_k^{-1})\left(Q(\overline{x}^k, y, z) - Q(x, \overline{y}^k, \overline{z}^k)\right)$$

$$- \sum_{k=0}^{K-1}\alpha_k^{-2}\left(Q(\overline{x}^{k+1}, y, z) - Q(x, \overline{y}^{k+1}, \overline{z}^{k+1})\right)$$

$$\overset{(a)}{\leq} \tau_x \alpha_0^{-2}\|x^0 - x\|^2 + \frac{1}{2\eta_y}\|y^0 - y\|^2 + \frac{1}{2\eta_z}\|\hat{z}^0 - z\|^2 + \frac{1}{2\eta_z}\|\eta_z^0 m^0\|_{\mathbf{P}}^2 + \frac{2nM^2 K}{\tau_x T}$$

$$- \frac{2\eta_y(K+2)^2}{9}\|\tilde{x}^K - x^{K-1}\|^2 + \frac{\eta_y(K+2)^2}{9}\|x^{K-1} - \tilde{x}^K\|^2 + \frac{1}{4\eta_y}\|y^K - y\|^2$$

$$+ \sum_{k=0}^{K-1}(\alpha_k^{-2} - \alpha_k^{-1})\left(Q(\overline{x}^k, y, z) - Q(x, \overline{y}^k, \overline{z}^k)\right)$$

$$- \sum_{k=0}^{K-1}\alpha_k^{-2}\left(Q(\overline{x}^{k+1}, y, z) - Q(x, \overline{y}^{k+1}, \overline{z}^{k+1})\right)$$

$$\overset{(b)}{=} \tau_x\|x^0 - x\|^2 + \frac{1}{2\eta_y}\|y^0 - y\|^2 + \frac{1}{2\eta_z}\|\hat{z}^0 - z\|^2 + \frac{1}{2\eta_z}\|\eta_z^0 m^0\|_{\mathbf{P}}^2 + \frac{2nM^2 K}{\tau_x T}$$

$$- \frac{\eta_y(K+2)^2}{9}\|\tilde{x}^K - x^{K-1}\|^2 + \frac{1}{4\eta_y}\|y^K - y\|^2 - \alpha_{K-1}^{-2}\left(Q(\overline{x}^K, y, z) - Q(x, \overline{y}^K, \overline{z}^K)\right)$$

$$+ \sum_{k=1}^{K-1}(\alpha_k^{-2} - \alpha_k^{-1} - \alpha_{k-1}^{-2})\left(Q(\overline{x}^k, y, z) - Q(x, \overline{y}^k, \overline{z}^k)\right)$$

$$\overset{(c)}{=} \tau_x\|x^0 - x\|^2 + \frac{1}{2\eta_y}\|y^0 - y\|^2 + \frac{1}{2\eta_z}\|\hat{z}^0 - z\|^2 + \frac{1}{2\eta_z}\|\eta_z^0 m^0\|_{\mathbf{P}}^2 + \frac{2nM^2 K}{\tau_x T}$$

$$- \frac{\eta_y(K+2)^2}{9}\|\tilde{x}^K - x^{K-1}\|^2 + \frac{1}{4\eta_y}\|y^K - y\|^2 - \sum_{k=1}^{K}\lambda_k\left(Q(\overline{x}^k, y, z) - Q(x, \overline{y}^k, \overline{z}^k)\right),$$

where (a) uses the Cauchy-Schwarz inequality, Young's inequality, and the initialization $\tilde{x}^0 = x^{-1}$ on line 1 of Algorithm 1; (b) uses the fact that $\alpha_0 = 1$, which follows from eq. (82); (c) uses the definition of $\lambda_k$ in eq. (85). Further, we obtain the following:

$$\tau_x \alpha_K^{-2}\|x^K - x\|^2 + \frac{1}{2\eta_y}\|y^K - y\|^2 + \frac{1}{2\eta_z}\|\hat{z}^K - z\|^2 + \frac{1}{2\eta_z}\|\eta_z^K m^K\|_{\mathbf{P}}^2$$

$$\overset{(a)}{\leq} \tau_x\|x^0 - x\|^2 + \frac{1}{2\eta_y}\|y^0 - y\|^2 + \frac{1}{2\eta_z}\|\hat{z}^0 - z\|^2 + \frac{1}{2\eta_z}\|\eta_z^0 m^0\|_{\mathbf{P}}^2 + \frac{2nM^2 K}{\tau_x T}$$

$$- \frac{\eta_y(K+2)^2}{9}\|\tilde{x}^K - x^{K-1}\|^2 + \frac{1}{4\eta_y}\|y^K - y\|^2 - \sum_{k=1}^{K}\lambda_k\left(Q(x_a^K, y, z) - Q(x, y_a^K, z_a^K)\right)$$

$$\overset{(b)}{=} \tau_x\|x^0 - x\|^2 + \frac{1}{2\eta_y}\|y^0 - y\|^2 + \frac{1}{2\eta_z}\|\hat{z}^0 - z\|^2 + \frac{1}{2\eta_z}\|\eta_z^0 m^0\|_{\mathbf{P}}^2 + \frac{2nM^2 K}{\tau_x T}$$

$$- \frac{\eta_y (K+2)^2}{9} \|\tilde{x}^K - x^{K-1}\|^2 + \frac{1}{4\eta_y} \|y^K - y\|^2 - \sum_{k=0}^{K-1} \alpha_k^{-1} \left( Q(x_a^K, y, z) - Q(x, y_a^K, z_a^K) \right),$$

where (a) uses the convexity of $Q(x, y, z)$ in $x$ (follows from Assumption 1) and the concavity of $Q(x, y, z)$ in $(y, z)$, line 14 of Algorithm 1, and the fact that $\lambda_k \geq 0$, which follows from eqs. (82) and (85); (b) use the definition of $\lambda_k$ in eq. (85) and the fact that $\alpha_0 = 1$, which follows from eq. (85). Next, we do rearranging and use eqs. (84) and (86), which gives the following:

$$\left( \sum_{k=0}^{K-1} \alpha_k^{-1} \right) \left( Q(x_a^K, y, z) - Q(x, y_a^K, z_a^K) \right) \leq \frac{r}{3} \|x\|^2 + \frac{6}{r} \|y\|^2 + \frac{15\chi^2}{r} \|z\|^2 + \frac{12nM^2K}{rT}.$$

Next, we divide both sides of the inequality by $\sum_{k=0}^{K-1} \alpha_k^{-1}$, which gives the following:

$$Q(x_a^K, y, z) - Q(x, y_a^K, z_a^K) \leq \left( \sum_{k=0}^{K-1} \alpha_k^{-1} \right)^{-1} \left( \frac{r}{3} \|x\|^2 + \frac{6}{r} \|y\|^2 + \frac{15\chi^2}{r} \|z\|^2 + \frac{12nM^2K}{rT} \right).$$

Further, we can estimate $\sum_{k=0}^{K-1} \alpha_k^{-1}$ as follows:

$$\sum_{k=0}^{K-1} \alpha_k^{-1} \overset{(a)}{=} \sum_{k=0}^{K-1} \frac{k+3}{3} = K + \frac{1}{3} \sum_{k=0}^{K-1} k = K + \frac{K(K-1)}{6} = \frac{K(K+5)}{6} \geq \frac{K^2}{6},$$

where (a) uses eq. (82). Plugging this into the previous inequality gives

$$Q(x_a^K, y, z) - Q(x, y_a^K, z_a^K) \leq \frac{6}{K^2} \left( \frac{r}{3} \|x\|^2 + \frac{6}{r} \|y\|^2 + \frac{15\chi^2}{r} \|z\|^2 + \frac{12nM^2K}{rT} \right)$$

$$= \frac{1}{K^2} \left( 2r\|x\|^2 + \frac{36}{r} \|y\|^2 + \frac{90\chi^2}{r} \|z\|^2 \right) + \frac{72nM^2}{rKT}$$

which concludes the proof. $\qquad \square$

