# OpenReview forum: "Lower Bounds and Optimal Algorithms for Non-Smooth Convex Decentralized Optimization over Time-Varying Networks"
_NeurIPS.cc/2024/Conference — NeurIPS 2024 poster_

### Official Review · Reviewer_3roP · 2024-07-07

**Soundness:** 4
**Presentation:** 4
**Contribution:** 3
**Rating:** 7
**Confidence:** 3

**Summary:**

This paper studies the problem of decentralized optimization for non-smooth convex opjectives and time-varying networks. The paper introduces an algorithm to solve this problem together with matching lower bounds on the required communication and subgradient computations, thereby proving that the proposed algorithm is optimal in this sense.

**Strengths:**

The paper is very well written and explains the concepts used in this work very clearly. Furthermore, the paper solves a relevant research question by proposing an optimal algorithm together with matching lower bounds. The results and proofs established in this paper seem to be correct.

**Weaknesses:**

In my experience, optimization algorithms can sometimes yield unsatisfactory performance despite having good theoretical guarantees. Therefore, I believe that adding (even a small) simulation example showcasing the proposed algorithm's performance on a relevant problem would greatly improve the paper.

Furthermore, while the paper is generally very well written, the algorithm could be explained a little bit better. Even just mentioning which lines of the algorithm correspond to which step in lines 282 - 296 would greatly help a reader who is not necessarily familiar with each of the references. In exchange, I believe that the justification for considering convex cost functions (Section 1.2) could be shortened.

Finally, there are a few small typos that should be corrected:
I think that $\mathbf{1}_p$ should be defined as $(1,\dots,1)^\top$; Algorithm 1 requires an initialization for $\bar{z}^0$ and $\bar{y}^0$, and in the proofs (e.g., after line 765) you need to define w.l.o.g. that $\tilde{x}^0 = x^{-1}$; in line 657, I think you should have $+\langle x_a^K,z \rangle$ and you need to flip the sign in (87); in lines 692-693, the function $F(x)$ maps from $(\mathbb{R}^n)^d$, i.e., the argument should be $w^*$ instead of $x^*$; in Appendix E.3, E.4, and E:5, you previously used the notation $W(k)$ instead of $W_k$; in the first equations of Appendix E.6 (after (b)), I believe it should be $y^k$ instead of $y^{k+1}$, which is corrected in the next equation, and I do not see why you need line 7 of Algorithm 1 for (d); after line 773, there should be $k$ instead of $K$ in the third sum.

**Questions:**

Why does (57) not change in Part(ii) (line 591)? I believe the result is correct, but I am not sure whether this statement is true.

**Limitations:**

I do not fully agree with the authors' answer to Question 2 in the NeurIPS Checklist, as limitations of the presented results and avenues for future research are not clearly discussed in the paper. More specifically, the authors only provide the required assumptions, but do not discuss whether these assumptions are restrictive in certain scenarios (except for Assumption 1) and whether these assumptions could be relaxed in future work.

---

> ### Author Rebuttal · Authors · 2024-08-06
>
> We thank the reviewer for the time, effort, and feedback. Below, we provide our detailed response to the review.
>
> ### Weaknesses
>
> >I believe that adding (even a small) simulation example showcasing the proposed algorithm's performance on a relevant problem would greatly improve the paper.
>
> Thank you for the suggestion. We are currently working on the basic experiments and will try to include them in the revised version of the paper.
>
> > Just mentioning which lines of the algorithm correspond to which step in lines 282 - 296 would greatly help a reader who is not necessarily familiar with each of the references.
>
> We will try to provide an additional explanation of the algorihtm development with references to the corresponding algorithmic lines in the revised version of the paper by shortening Section 1.2, or by using an additional page if the paper is accepted.
>
> >Finally, there are a few small typos that should be corrected.
>
> Thank you for finding the typos. We have corrected them as follows:
> - changed $\mathbb{1}_p = (1,\ldots,1)^\top$;
> - added missing initializations on line 1 of Algorithm 1:
>   - added explicit initialization $\overline{y}^0 = y^0$ and $\overline{z}^0 = z^0$ (note that $\overline{y}^0$ and $\overline{z}^0$ are not used eventually due to line 4 and the fact that we use $\alpha_0 = 1$);
>   - added explicit initialization $x^{-1} = \tilde{x}^0 = x^0$ and annotated corresponding transitions after line 765;
> - fixed the sign of $\langle x_a^K,z \rangle$ on line 657 and after, and fixed the sign in equation (87);
> - replaced $x^*$ with $w^*$ on lines 692-693;
> - we use the notation $\mathbf{W}_k$ when $k$ is the iteration number, and $\mathbf{W}(\tau)$ when $\tau$ is the continuous time, which is mentioned on line 6 of Algorithm 1;
> - in the first equation of Appendix E.6:
>   - $y^{k+1}$ after (b) is a typo and is replaced with $y^k$;
>   - (d) indeed does not use line 7 of Algorithm 1, it is just an algebraic transformation;
> - fixed the third sum after line 773.
>
> ### Questions
>
> >Why does (57) not change in Part(ii) (line 591)? I believe the result is correct, but I am not sure whether this statement is true.
>
> There is indeed a small inaccuracy on line 591. Similarly to part(i) and due to Lemma 2 equation (57) will change to the following:
> $$
> \\mathcal{M}_i^{\\text{sub}}(\\tau)
> \\subset
> \\begin{cases}
>     \\mathcal{K}\_{2p + 2} & i \\in \\mathcal{V}_1 \\text{ or } \\left(i \\in \\mathcal{V}_3 \\text{ and } i \\leq 2n/3 + q + 1\\right) \\\\
>     \\mathcal{K}\_{2p + 1} & i \\in \\mathcal{V}_2 \\text{ or } \\left(i \\in \\mathcal{V}_3 \\text{ and } i > 2n/3 + q + 1\\right)
> \\end{cases},
> $$
> and the rest of the proof remains unchanged, including the last step:
> $$
> \\begin{aligned}
> \\mathcal{M}_i(\\tau)
> &\\subset
> \\mathcal{M}_i^{\\text{sub}}(\\tau) \\cup \\mathcal{M}_i^{\\text{com}}(\\tau)
> \\\\&\\subset
> \\begin{cases}
>     \\mathcal{K}\_{2p + 2} & i \\in \\mathcal{V}_1 \\text{ or } \\left(i \\in \\mathcal{V}_3 \\text{ and } i \\leq 2n/3 + q + 1\\right) \\\\
>     \\mathcal{K}\_{2p + 1} & i \\in \\mathcal{V}_2 \\text{ or } \\left(i \\in \\mathcal{V}_3 \\text{ and } i > 2n/3 + q + 1\\right)
> \\end{cases}.
> \\end{aligned}
> $$
> We added this clarification to the proof. Thank you for noticing this.
>
> ### Limitations
>
> >More specifically, the authors only provide the required assumptions, but do not discuss whether these assumptions are restrictive in certain scenarios (except for Assumption 1) and whether these assumptions could be relaxed in future work.
>
> We will try to add an appropriate discussion on the restrictiveness and the possibility of relaxing the assumptions. For instance, Assumption 5 can be relaxed to a more general joint spectrum property (Nedic et al., 2017), and a corresponding extension of our results would be trivial. Assumption 2 can probably be relaxed too, if we replace the subgradient method with some adaptive method like Adagrad [1].
>
> [1] Duchi, John, Elad Hazan, and Yoram Singer. "Adaptive subgradient methods for online learning and stochastic optimization." Journal of machine learning research 12.7 (2011).

---

> > ### Comment · Reviewer_3roP · 2024-08-09
> >
> > Thank you for your clarifications. I read them carefully and decided to keep my original score, which is already relatively high.

---

### Official Review · Reviewer_ap5m · 2024-07-13

**Soundness:** 3
**Presentation:** 3
**Contribution:** 3
**Rating:** 7
**Confidence:** 2

**Summary:**

The authors introduce an algorithm which optimally bounds the complexity of algorithms for non-smooth convex decentralised optimisation over time varying networks.

**Strengths:**

The paper introduces an algorithm for an unsolved problem setting as there have been several prior works that provide solutions and bounds for the smooth convex case but this is the first work that that provides bounds an optimal algorithms for the non-smooth case in a time varying domain.

**Weaknesses:**

The paper can be quite dense and while the authors provide slight intuition about the proof sketch in the main paper, most of the actual paper lies in the appendix which can make it quite inaccessible.

**Questions:**

Based on the above comments, I would ask the authors the following question,

 - Would it be possible to make the main paper less dense and focus more on guiding the user through the intuition of the proofs?

**Limitations:**

The authors have sufficiently addressed any limitations that might arise from the submitted work.

---

> ### Author Rebuttal · Authors · 2024-08-06
>
> We thank the reviewer for the time, effort, and feedback. Below, we provide our detailed response to the review.
>
> ### Weaknesses and Questions
>
> >The paper can be quite dense and while the authors provide slight intuition about the proof sketch in the main paper, most of the actual paper lies in the appendix which can make it quite inaccessible.
>
> >Would it be possible to make the main paper less dense and focus more on guiding the user through the intuition of the proofs?
>
> Thank you for the suggestion. For the proof of lower bounds, the main theoretical ideas are listed on lines 204-223. We will try to enrich this part by adding a more detailed explanation of how to "assemble" these ideas into the final proof in the revised version of the paper.
> For the proof of algorithm convergence, the main algorithmic components are listed on lines 282-296. We will try to provide an informal explanation of how these components work from a theoretical perspective and how to obtain the final proof using them, with appropriate references to the lemmas in the appendix.
> We will try to obtain additional space for these changes, perhaps by shortening some parts of the introduction (as suggested by Reviewer 3roP), or by using an extra page if the paper is accepted.

---

> > ### Comment · Reviewer_ap5m · 2024-08-09
> > **Read your rebuttal**
> >
> > I thank the authors for their clarifications. Based on all the information presented I will be keeping my score as-is. Best of luck to the authors for the final evaluation.

---

### Official Review · Reviewer_Eyy5 · 2024-07-13

**Soundness:** 3
**Presentation:** 3
**Contribution:** 3
**Rating:** 7
**Confidence:** 2

**Summary:**

The paper studies non-smooth decentralized optimization with time-varying communication networks. The paper presents execution time lower bounds of subgradient algorithms for strongly-convex and convex cases. Then, the paper develops algorithms achieving matching execution time.

**Strengths:**

I am not very familiar with the field of this paper. As far as I see, the paper makes good theoretical contributions by resolving the open problem concerning the complexity of non-smooth decentralized optimization with time-varying communication networks. This problem is natural and has many potential applications in practice. The auithors draw detailed comparison with existing works and present their results well. There is no major flaw I see in this work, but again I am not very familiar with the field. One minor weakness might be a lack of numerical experiments validating the convergence rate of the proposed algorithm. Nonetheless, I think it is not a big issue for a theory paper.

**Weaknesses:**

There is no major flaw I see in this work, but again I am not very familiar with the field. One minor weakness might be a lack of numerical experiments validating the convergence rate of the proposed algorithm. Nonetheless, I think it is not a big issue for a theory paper.

**Questions:**

Why does the objective in eq (1) have a quadratic regularization?

The theorem 1 and 2 state lower bounds of execution times but the authors interpret these results as communication and computation bounds. I wonder can you make this argument formal in the classical definition of communication complexity (Yao, 1979) and computation complexity (Papadimitriou, 2003).

References:

Yao, A. C. C. (1979). Some complexity questions related to distributive computing (preliminary report). In Proceedings of the eleventh annual ACM symposium on Theory of computing (pp. 209-213).

Papadimitriou, C. H. (2003). Computational complexity. In Encyclopedia of computer science (pp. 260-265).

**Limitations:**

Yes.

---

> ### Author Rebuttal · Authors · 2024-08-06
>
> We thank the reviewer for the time, effort, and feedback. Below, we provide our detailed response to the review.
>
> ### Weaknesses
>
> >One minor weakness might be a lack of numerical experiments validating the convergence rate of the proposed algorithm. Nonetheless, I think it is not a big issue for a theory paper.
>
> Thank you for the suggestion. We are currently working on the basic experiments and will try to include them in the revised version of the paper.
>
> ### Questions
>
> >Why does the objective in eq (1) have a quadratic regularization?
>
> We decided to use an explicit regularizer because it allows for a cleaner transition from the original problem (1) to the saddle-point reformulation (14). Note that any strongly convex finite-sum optimization problem can be presented in the form of equation (1). In the case of a non-strongly convex problem, we use the standard regularization technique, which allows the reduction of convex problems to strongly convex ones (see, for instance, [1]). We describe this in Appendix D.3. We will try to add this clarification to the main part of the paper in the revised version.
>
> [1] Allen-Zhu, Zeyuan, and Elad Hazan. "Optimal black-box reductions between optimization objectives." Advances in Neural Information Processing Systems 29 (2016).
>
> >The theorem 1 and 2 state lower bounds of execution times but the authors interpret these results as communication and computation bounds. I wonder can you make this argument formal in the classical definition of communication complexity (Yao, 1979) and computation complexity (Papadimitriou, 2003).
>
> Unfortunately, to the best of our knowledge, the vast majority of results in the optimization literature are provided in terms of limited computation models, for instance, black-box optimization procedures [2] (similar to what we use in the paper), or polynomial time methods [3]. Note that the latter is significantly different from the classical notion of polynomial time algorithms. The important aspect is that we need to obtain very refined results, rather than classical polynomial or exponential complexity, which is, in some sense, similar to the complexity results obtained for sorting algorithms in the comparison tree model, or for matrix multiplication algorithms in terms of algebraic complexity. Another potential issue is that our results depend not on the size of the input, but rather on the internal properties of the problem such as Lipschitz constants, network condition numbers, etc. Thus, making the arguments of Theorems 1 and 2 in terms of classical computation models may be difficult, at least within the limited author response period. However, this would be an interesting direction for future research.
>
> [2] Nesterov, Yu E. "Introductory Lectures on Convex Optimization. A Basic Course." (2004).
>
> [3] Nemirovski, Arkadi. "Interior point polynomial time methods in convex programming." Lecture notes 42.16 (2004): 3215-3224.

---

> > ### Comment · Reviewer_Eyy5 · 2024-08-08
> >
> > Thank you for the clarification, I decide to keep my positive rating.

---

### Official Review · Reviewer_7QFG · 2024-07-15

**Soundness:** 4
**Presentation:** 4
**Contribution:** 4
**Rating:** 8
**Confidence:** 3

**Summary:**

This paper derives lower bounds on the communication and computation complexities of solving non-smooth convex decentralized optimization problems over time-varying networks and designs and algorithm that achieves these lower bounds.

**Strengths:**

The problem studied in the paper is interesting.

The results look to be rigorous and strong (by achieving the derived lower bound).

**Weaknesses:**

None.

**Questions:**

None.

**Limitations:**

Yes.

---

> ### Author Rebuttal · Authors · 2024-08-06
>
> We thank the reviewer for the time, effort, and appreciation of the theoretical results of our paper.

---

### Author Rebuttal · Authors · 2024-08-06

We thank the reviewers for their time, effort, and high evaluation of our work. As far as we understand, there were no major common issues raised by the reviewers. Hence, we are providing our detailed responses to each review in separate messages.

---

### Decision · Program_Chairs · 2024-09-25

**Decision:**

Accept (poster)

**Comment:**

This paper derives lower bounds on the communication and computation complexities of solving non-smooth convex decentralized optimization problems over time-varying networks and designs an algorithm that achieves these lower bounds. All reviewers appreciate the contributions of this work, recognizing its novelty and nontrivial nature. The only minor weakness lies in the somewhat limited simulation setups.

In the final version I recommend the authors to discuss the time-varying assumption. In general, time-varying graphs are assumed to be contractive over a finite length, but this work seem to assume contraction for every graph.